# Evidence for wastewaters as environments where mobile antibiotic resistance genes emerge

Fanny Berglund [1,2], Stefan Ebmeyer[1,2], Erik Kristiansson[2,3] & D. G. Joakim Larsson [1,2 ✉]

The emergence and spread of mobile antibiotic resistance genes (ARGs) in pathogens have become a serious threat to global health. Still little is known about where ARGs gain mobility in the first place. Here, we aimed to collect evidence indicating where such initial mobilization events of clinically relevant ARGs may have occurred. We found that the majority of previously identified origin species did not carry the mobilizing elements that likely enabled intracellular mobility of the ARGs, suggesting a necessary interplay between different bacteria. Analyses of a broad range of metagenomes revealed that wastewaters and wastewater-impacted environments had by far the highest abundance of both origin species and corresponding mobilizing elements. Most origin species were only occasionally detected in other environments. Co-occurrence of origin species and corresponding mobilizing elements were rare in human microbiota. Our results identify wastewaters and wastewater-impacted environments as plausible arenas for the initial mobilization of resistance genes.

[1] Department of Infectious Diseases, Institute of Biomedicine, the Sahlgrenska Academy, University of Gothenburg, Gothenburg, Sweden. [2] Centre for Antibiotic Resistance Research in Gothenburg (CARe), University of Gothenburg, Gothenburg, Sweden. [3] Department of Mathematical Sciences, Chalmers University of Technology and University of Gothenburg, Gothenburg, Sweden. ✉email: joakim.larsson@fysiologi.gu.se

Antibiotic resistance has become a major global threat to public health[1]. While some species are intrinsically resistant to certain antibiotics, many pathogens have become resistant as a result of mutations in pre-existing DNA or through the acquisition of mobile antibiotic resistance genes (ARGs) via horizontal gene transfer[2,3]. While most ARGs are located in genetic contexts with limited possibilities to become transferred to pathogens, many ARGs have increased their mobility potential (here referred to as mobilization) by association with small transposable elements, such as insertion sequences (ISs). Many of these ARGs have then spread horizontally to pathogens through association with other mobile genetic elements, such as plasmids or transposons[4]. Once mobilized, an ARG may spread fast, particularly when promoted by selection pressure from antibiotics[5].

Overall, our knowledge about the mobilization process, including in which environments ARGs are initially mobilized, is limited[6–8]. It has long been hypothesized that many ARGs that today are clinically relevant have their origin in environmental bacteria[9,10]. Indeed, environmental and commensal bacterial communities harbor a vast diversity of ARGs, many of which have not yet been detected in pathogens[11–14]. Lately, more efforts have been dedicated to identifying the origin species in which many of the today frequently encountered mobile ARGs gained mobility. In this context, origin species therefore describes the species where the evolutionary most recent ancestor gene of a today mobile ARG (≥95% sequence identity) is widespread, but not commonly associated with any of the mobile genetic element(s) that were likely involved in the transition of the ARG to its clinically relevant context[15]. A newly published systematic summary of the current literature outlined the recent evolutionary origins of around 40 mobile ARGs, originating from a little more than 30 different taxa[15]. Here, many of the ISs likely involved in the mobilization process, hereafter referred to as mobilizing insertion sequences (MISE), were also summarized. All identified origin species belonged to the phylum Proteobacteria, possibly reflecting that the majority of the investigated ARGs were beta-lactamases. In all but one case, the origin was a bacterial species that at least occasionally is associated with infections in humans or domestic animals. This observation would speak in favor of the human/domestic animal microbiota being the most critically important site for mobilization and transfer to pathogens, possibly under selection pressure from antibiotics. On the other hand, for ~95% of all known mobile ARGs, the origin is still unknown, pointing toward either yet unsequenced bacterial species primarily thriving in the external environment as the likely origins, or that the mobilization event happened so far back in time that the ancestor has evolved to the point that there is no identifiable recent origin.

Selection pressure from antibiotics can promote mobilization in multiple steps of the process[8]. First, it may enrich the abundance and diversity of not yet mobilized ARGs[16,17]. Second, such exposure may favor the capture of previously immobile ARGs by mobilizing genetic elements such as ISs or insertion sequence common regions (ISCRs)[18,19]. Such co-localization has in several cases been shown to increase expression of the ARG, either through IS-internal promoters, creation of hybrid-promoters or the disruption of repressors[20,21]. In addition, through the co-localization with e.g., an IS element, the ability to move intra-cellularly to e.g., plasmids, particularly multi-copy-plasmids, could similarly lead to increased expression[22–24]. Indeed, co-localization with IS elements are important drivers of horizontal transfer of ARGs, and many mobile ARGs are strongly associated with certain IS elements[22,25,26]. Third, it may promote the rate of horizontal gene transfer, as is the case with many different stressors, giving the newly mobilized ARG more opportunities to reach other hosts[18,19,24]. Still, the most critical aspect is probably the strong advantage the exposure to an antibiotic provides to a bacterium that has acquired an ARG via horizontal gene transfer but belongs to a species that otherwise is sensitive to the antibiotic in question[8]. Taken together, this points to a higher risk of emergence of new mobile ARGs in environments that regularly are exposed to antibiotics, such as the human or domestic animal microbiota, or environments polluted with antibiotics, such as different wastewaters.

While almost all known origin species at least occasionally are associated with the human/domestic animal microbiota[15], knowledge of their wider environmental distribution is largely lacking. Today, we have access to an unprecedented amount of DNA sequencing data, enabling overarching analysis of the bacterial communities in numerous environments. An analysis of the presence and abundance of known origin species in a wide set of ecosystems could add to our understanding of what have been likely environments where these ARGs were initially mobilized. This could be further strengthened by an analysis of the distribution of mobilizing agents, particularly those suspected to have been involved in the mobilization of ARGs (the MISE). By using the abundance of crAssphage as an indicator of environments subjected to fecal pollution, environments having a possible connectivity to the human microbiome could also be identified[27,28]. If clear patterns would arise from such an analysis, it is plausible that those environments could have been a place where mobilization events have occurred in the past and could therefore also be likely arenas for future mobilization events. Thereby, this knowledge could ultimately inform actions to mitigate risks for the emergence of new mobile ARGs in pathogens.

In this study, we aimed to collect and compare evidence that could identify environments where the initial mobilization of ARGs with previously described origin species were most likely to have taken place. When studying the genomes of the investigated origin species, we found that only a minority of origin species carried MISE associated with the mobilization of the corresponding ARG. This indicates that in many cases the intracellular mobilization may have required the acquisition of MISE from another bacterial strain or species, either directly or indirectly through free DNA. By using a database optimized for the purpose of discriminating reads from 22 origin species from that of other species, we analyzed a large set of metagenomic data from different environment types for the presence and abundance of origin species as well as for their corresponding MISE. We show that the studied origin species are particularly abundant in wastewaters and wastewater-impacted environments. In addition, origin species and their corresponding MISE coexisted in a large majority of the wastewater environments, which was not the case in other environments, including the human microbiome. Taken together, our analysis points toward wastewaters and wastewater treatment plants (WWTPs) as environments in which mobilization and transfer events have likely occurred in the past. These environments could very well be arenas for similar future events.

## Results

### Implementation of a method for identifying origin species in metagenomes.
It has previously been established that certain mobile ARGs originate from the chromosome of certain bacterial species. A recent study by Ebmeyer et al. summarized the current knowledge regarding the taxonomic origins of mobile ARGs and proposed criteria for the identification of such origin species based on the collective knowledge within the field[15]. Therefore, 22 origin species outlined by Ebmeyer et al. were investigated in this study (Table 1). In order to accurately quantify the presence of

**Table 1 The investigated origin species together with the respective mobilized antibiotic resistance gene (ARG) and associated IS/ISCR-element (MISE).**

| Origin species | ARG | Antibiotic class | Associated IS/ISCR |
|---|---|---|---|
| *Acinetobacter baumannii* | OXA-51/345-like | Beta-lactam | ISAba1 |
| *Acinetobacter guillouiae* | APH(3')-IV | Aminoglycosides | ISAba125, ISAba14 |
| *Acinetobacter radioresistens* | OXA-23 | Beta-lactam | ISAba1, ISAba4 |
| *Aeromonas allosaccharophila* | FOX | Beta-lactam | IS26, ISAs2, Tn3-like |
| *Aeromonas caviae* | MOX-2 | Beta-lactam | ISKpn9 |
| *Aeromonas media* | MOX-9 | Beta-lactam | ISKpn9 |
| *Aeromonas sanarellii* | CMY-1/MOX-1 | Beta-lactam | ISCR1 |
| *Citrobacter freundii complex* | CMY-2-like | Beta-lactam | ISEcp1 |
| *Citrobacter freundii* | qnrB | Fluoroquinolones | ISCR1, ISEcp1, IS3000, IS6100, IS26 |
| *Enterobacter asburiae* | ACT-1 | Beta-lactam | Unknown |
| *Enterobacter cloacae* | FosA1 | Fosfomycin | Tn2921, IS4 |
| *Enterobacter cloacae* | MIR-1 | Beta-lactam | ISPps1 |
| *Enterobacter mori* | QnrE | Fluoroquinolones | ISEcp1 |
| *Hafnia alvei/paralvei* | ACC | Beta-lactam | ISEcp1 |
| *Klebsiella pneumoniae* | SHV | Beta-lactam | IS26, IS102 |
| *Klebsiella pneumoniae* | FosA5/6 | Fosfomycin | IS10, IS1, IS26 |
| *Klebsiella pneumoniae* | OqxAB | Fluoroquinolones | IS26 |
| *Kluyvera ascorbata* | CTX-M-1,2,3,4,5,6,7 | Beta-lactam | ISEcp1, ISCR1 |
| *Kluyvera georgiana* | CTX-M-8/9/25 | Beta-lactam | IS10, ISEcp1, ISCR1 |
| *Kluyvera georgiana* | FosA3/4 | Fosfomycin | IS26, ISEcp1 |
| *Leclercia adecarboxylata* | FosA8 | Fosfomycin | Unknown |
| *Moraxella pluranimalium* | MCR-2 | Colistin | IS1595 |
| *Morganella morganii* | DHA | Beta-lactam | Unknown |
| *Rheinheimera pacifica* | LMB-1 | Beta-lactam | IS6, IS91 |
| *Shewanella algae* | QnrA | Fluoroquinolones | ISCR1 |
| *Shewanella xianamenensis* | OXA-48/181 | Beta-lactam | ISEcp1 |

these origin species, a method based on Kraken2 was developed. The method uses a k-mer based approach to match reads directly to a specially developed and optimized database. In short, the database was initially created using the complete Kraken2 database complemented with manually curated genomes from the investigated origin species and their closely related genera (Supplementary Data 1). Based on validation of the method, the parameters of Kraken2 were set to limit the false positive rate (FPR) while allowing a rather large proportion of fragments to be unclassified at the species level, and thus discarded. The average true positive rate (TPR) of the method to correctly identify origin species was estimated to be 0.95 (Supplementary Table 1) while the FPR was estimated to $9.59 \times 10^{-7}$ (Supplementary Table 2). See Methods and Supplementary Tables 1–3 for more information about the database creation and evaluation.

**Origin species were mainly found in wastewaters and human stool.** Next, we applied the method to shotgun metagenomic data from a large range of environments (Table 2 and Supplementary Table 4). In summary, the data came from 30 different datasets and was comprised of 2496 unique samples, corresponding to ~203 billion reads (~22 trillion bases). The data could be divided into five main categories: human (seven datasets), animal (four datasets), soil (four datasets), water/sediment (nine datasets) and wastewater treatment plants (WWTP, six datasets). The microbial composition of the samples was dominated by Proteobacteria for WWTP and Water/Sediments data, most of the soil environments were dominated by either Actinobacteria or Proteobacteria while the human and animal related environments consisted of a mixture of mainly Firmicutes, Bacteroidetes and Actinobacteria (Supplementary Fig. 1). The two Indian datasets (upstream PETL and Kazipally lake) had among the higher relative abundances for most origin species (Fig. 1a, Supplementary Fig. 2a and Supplementary Data 2). Both of these represent aquatic environments polluted with antibiotics from

drug manufacturing[29]. Excluding the industrially polluted environments, the WWTP environments, especially influents, together with an Indian river (Pune river) polluted with urban waste, harbored the highest relative abundance of origin species. (Fig. 1a, Supplementary Fig. 2b and Supplementary Data 2). In fact, almost all origin species were present in the majority of all samples of the WWTP influent (102 in total) from all regions (Fig. 1b and Supplementary Data 2). The clear exceptions were *Shewanella algae* and *Rheinheimera pacifica* which were not common in influents, and to some extent *Moraxella pluranimilum*, *Kluyvera georgiana* and *Hafnia alvei* that showed some regional differences.

Although the relative abundance of all origin species combined was the highest in the polluted waters and WWTP environments, there were some differences in the abundance of individual species (Fig. 1a, Supplementary Fig. 3 and Supplementary Data 2). Interestingly, only *Hafnia paralvei*, *Rheinheimera pacifica* and *Shewanella algae* were, on average, more abundant in environments other than WWTPs and polluted waters (Supplementary Table 5). There were also environments in which the abundance of origin species varied considerably between samples. The species *Citrobacter freundii*, *Enterobacter asburiae*, *Enterobacter mori*, *Hafnia alvei*, *Klebsiella pneumonia*, and *Morganella morganii* were present in several samples in the human stool data with a far greater relative abundance than in any other environment, (Supplementary Fig. 4) while they were undetectable or just above the detection limit in the majority of samples. Furthermore, human stool samples taken from subjects treated with antibiotics ($n = 81$) had a significantly higher abundance of *Enterobacter mori* compared to the subjects not treated with antibiotics ($n = 303$)(Wilcoxon-Mann-Whitney test, $p = 0.049$). In addition, there were large differences between stool samples taken from different countries, where the abundance of some origin species was high in certain countries while being very low or completely undetectable in others (Supplementary Fig. 5). Still,

**Table 2 The analyzed datasets.**

| Dataset source/name | Type | Location | Samples | Size (reads) | Size (Mbases) |
|---|---|---|---|---|---|
| *Human* | | | | | |
| HMP | Various body sites | USA | 756 | $3.66 \times 10^{10}$ | 3119829 |
| PD gut | Stool | Germany | 58 | $2.76 \times 10^{9}$ | 273574 |
| Antibiotic treated human | Stool | Denmark | 57 | $4.02 \times 10^{9}$ | 375588 |
| Ceph. treated human | Stool | Canada | 72 | $9.52 \times 10^{9}$ | 962015 |
| Hadza hunter | Stool | Tanzania + Italy | 38 | $8.29 \times 10^{8}$ | 78617 |
| DB gut | Stool | China | 114 | $3.16 \times 10^{9}$ | 237079 |
| Italian youth | Stool | Italy | 6 | $2.12 \times 10^{8}$ | 26449 |
| *Water/Sediment* | | | | | |
| Aquaculture | Aquaculture (water from various sites) | China | 13 | $9.35 \times 10^{8}$ | 140234 |
| Pune river | Polluted river sediments | India | 10 | $2.8 \times 10^{9}$ | 353169 |
| Oil spill | Ocean deep water | Gulf of Mexico | 14 | $3.51 \times 10^{9}$ | 354812 |
| Tara Ocean | Ocean deep + surface water | Multiple locations | 135 | $4.95 \times 10^{10}$ | 4891581 |
| Indian lake | Polluted lake sediment | India | 1 | $6.69 \times 10^{7}$ | 6760 |
| Daisy lake | Lake sediment | Canada | 25 | $2.97 \times 10^{8}$ | 39822 |
| Sudbury lake | Lake sediment | Canada | 120 | $4.9 \times 10^{8}$ | 67534 |
| Salt marsh | Salt marsh sediment | UK | 38 | $6.5 \times 10^{8}$ | 98059 |
| *WWTP* | | | | | |
| Global sewage | Influent (untreated urban sewage) | Multiple countries | 80 | $9.58 \times 10^{9}$ | 1447288 |
| Swedish WWTP | Various sites | Sweden | 70 | $5.18 \times 10^{9}$ | 482405 |
| Korea WWTP | Various sites | Korea | 23 | $2.51 \times 10^{9}$ | 379612 |
| Hospital effluent UK | Untreated hospital effluent | UK | 8 | $5.83 \times 10^{8}$ | 87475 |
| Hospital effluent India | Untreated hospital effluent | India | 1 | $2.8 \times 10^{9}$ | 353170 |
| Activated sludge 1 | Activated sludge | Denmark | 4 | $1.01 \times 10^{9}$ | 150822 |
| Activated sludge 2 | Activated sludge | Argentina | 12 | $6.39 \times 10^{8}$ | 95830 |
| *Soil* | | | | | |
| Agriculture soil | Agriculture soil | Finland | 18 | $9.94 \times 10^{8}$ | 150072 |
| Forest soil | Forest soil | North America | 21 | $2.75 \times 10^{9}$ | 206030 |
| Desert | Desert rocks | USA | 6 | $8.69 \times 10^{8}$ | 216604 |
| Grassland | Partially polluted soil | Argentina | 12 | $1.15 \times 10^{8}$ | 11600 |
| *Animal* | | | | | |
| Pig gut | Feces pig | Multiple countries | 295 | $1.86 \times 10^{10}$ | 1742883 |
| Pig and poultry | Feces pig and poultry | Multiple countries | 387 | $3.35 \times 10^{10}$ | 4998116 |
| Cow feces | Feces | USA | 14 | $6.47 \times 10^{8}$ | 92199 |
| Cow rumen | Rumen | France | 82 | $9.54 \times 10^{9}$ | 835536 |
| *Total* | | | 2496 | $2.03 \times 10^{11}$ | 22274764 |

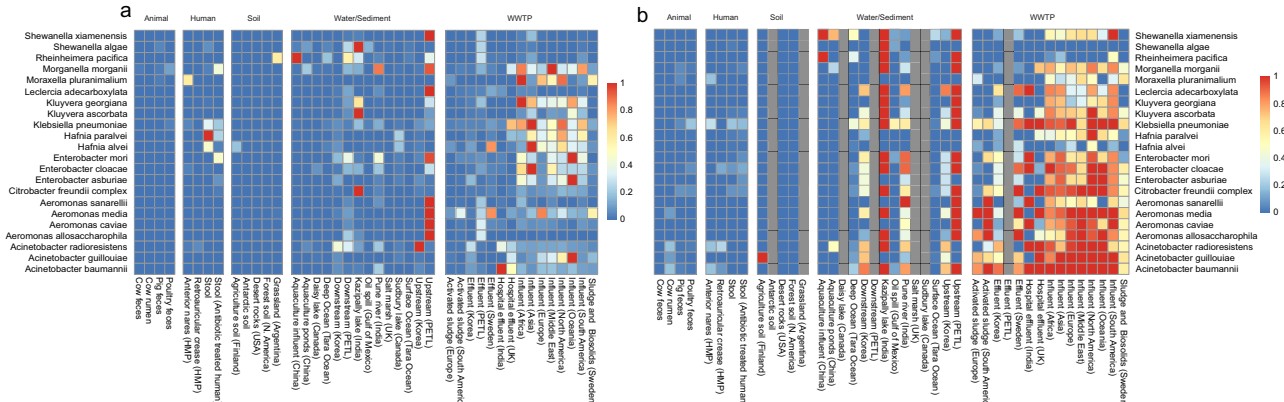

**Fig. 1 Relative abundance and prevalence of 22 known origin species.** Average relative abundance of the 22 investigated origin species in different environments (2496 samples) normalized by each species' highest relative abundance (**a**), and the prevalence of the 22 investigated origin species as fraction of samples with detectable levels of the corresponding species within each environment type (**b**). Some environments are excluded from the analysis in (**b**) due to too few identified bacterial fragments.

no origin species was significantly more abundant in human stool than in any other environment (Wilcoxon-Mann-Whitney test) and all species but *Shewanella algae* were significantly more likely to be detected in WWTP influent samples ($n = 102$) than in stool samples ($n = 384$)(Fisher's exact test, odds-ratios > 19, adjusted $p$ values < 0.001, Supplementary Table 6). In addition,

*Acinetobacter radioresistens* and *Moraxella pluranimalium* had a high abundance in a small fraction of samples taken from human anterior nares (nose). Still, the large majority of the human samples had a low relative abundance of the origin species, resulting in the low averages seen in Fig. 1a and Supplementary Figs. 2 and 3.

Interestingly, none of the soil-associated environments harbored any origin species in high abundance relative to the other environments, except for *Hafnia alvei* detected in agricultural soil from Finland and *Rheinheimera pacifica* found in Argentinian grassland. A similar pattern was found for the animal-associated data, where only *Morganella morganii* (in cow, pig and poultry feces), *Moraxella pluranimum* (pig feces), *Citrobacter freundii* complex (pig and poultry feces) and *Shewanella algae* (in pig feces) had detectable levels, although considerably lower than in other environments. In the data from water/sediments, the samples from the polluted environments Kazipally lake, downstream PETL and upstream PETL together with the samples from rivers downstream of WWTPs in Korea (Downstream Korea) and Pune river contained the most origin species, and were more resembling the WWTP influent than the other water-related samples. Possibly this could reflect input from untreated sewage in some, but not all, cases[27]. Excluding those environments, the origin species that were most abundant in water samples were *Shewanella algae* (mainly found in marine samples), *Rheinheimera pacifica* (most abundant in the influent of two aquaculture sites in China) and *Acinetobacter radioresistens* (upstream from a WWTP in Korea).

### Mobilizing IS-elements in genomes of origin species.

The majority of the ARGs for which the origin species have been identified are hypothesized to have been mobilized with the help of one or several IS elements[15]. However, it is not known if the mobilizing IS elements (MISE) were already present in the genome of the origin species before mobilization or if they had been acquired from elsewhere. Therefore, to investigate if the selected origin species regularly are carriers of the MISE, we analyzed all available complete genomes of the origin species (Supplemental Data 1). The results, visualized as the fraction of genomes carrying the MISE (Fig. 2 and Supplementary Data 2), showed that only a few of the origin species were carrying the MISE suspected to have previously been involved in the mobilization of an ARG from the corresponding species. There were, however, a few exceptions; both *Aeromonas media* and *Aeromonas caviae* were shown to sometimes carry ISKpn3 (associated with the mobilization of blaMOX-2 and blaMOX-9) in their genome, *Aeromonas allosacharophila* carried ISAs2 (blaFOX), *Acinetobacter*

*baumannii* carried ISAba1 (blaOXA-51/345-like), *Klebsiella pneumoniae* carried IS1 and IS26 (FosA5/6 and blaSHV) and *Citrobacter freundii* carried IS26 (qnrB). Furthermore, some MISE were found in multiple origin species, both in chromosomes and on plasmids, while some MISE were not found in any of the origin species. An analysis of 28 740 complete genomes (containing 7283 unique species) from NCBI RefSeq (downloaded September 2022) revealed that among the 223 unique bacterial species carrying MISE, 203 belonged to the phylum Proteobacteria (Fisher's exact test, p-value < 0.001). Furthermore, when comparing the taxonomical lineages of the unique species carrying MISE with the taxonomical lineage of the investigated origin species, we found that the large majority of MISE were present in species often belonging to the same genus or family as the species in which they have been associated with mobilization (Supplementary Fig. 6). Thus, the results indicate that many of the mobilization events likely required the MISE to be provided from the surrounding environment to the species originally carrying the not yet mobilized ARG. Here, the MISE could have been obtained from another strain or species, probably closely related to the origin species, or in the form of extracellular DNA from such strains/species.

### IS-elements in the studied environments.

Since many of the origin species did not carry the MISE in their genome, mobilization would have to require an interplay between different bacteria. Therefore, to investigate in what environments these MISE are present and abundant, we searched all metagenomic data for all known IS elements (Fig. 3 and Supplementary Data 2). Although the highest relative abundance of all IS elements was found in poultry feces (Fig. 3b), we found that the average relative abundance of the MISE was the highest in untreated hospital effluent and WWTP influent, together with poultry feces (Fig. 3a). In fact, all but one of the studied MISE (IS10R) were more abundant in the WWTP influent (n = 102) compared to all other environments (Wilcoxon-Mann-Whitney test, p-value < 0.001) as well as more likely to be above the detection limit (Fisher test, p < 0.001). As only a minority of the origin species were shown to carry their corresponding MISE, the high abundance of MISE in WWTP was probably not a result of the high abundance of origin species.

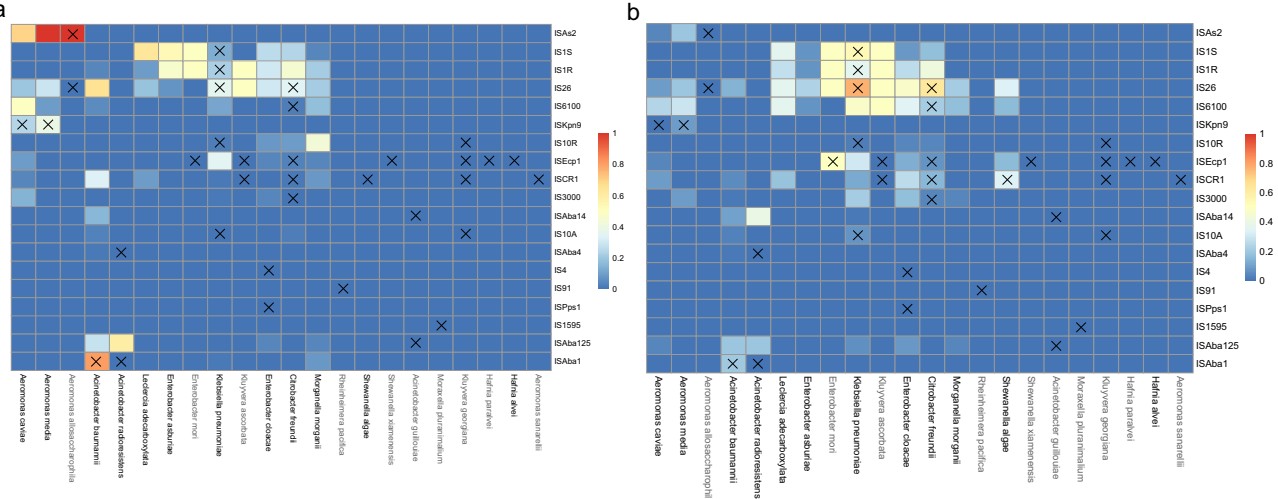

**Fig. 2 Origin species' genomes carrying IS elements associated with mobilization from the corresponding species.** Fraction of genomes (chromosome in (**a**) and plasmid in (**b**)) carrying any of the IS elements showed to have been involved in mobilization of an antibiotic resistance gene (ARG) from the studied origin species. A cross indicates that the IS has likely been involved in the mobilization of an ARG from the corresponding species. A gray color of the species name indicates that there were fewer than five genomes used in the analysis.

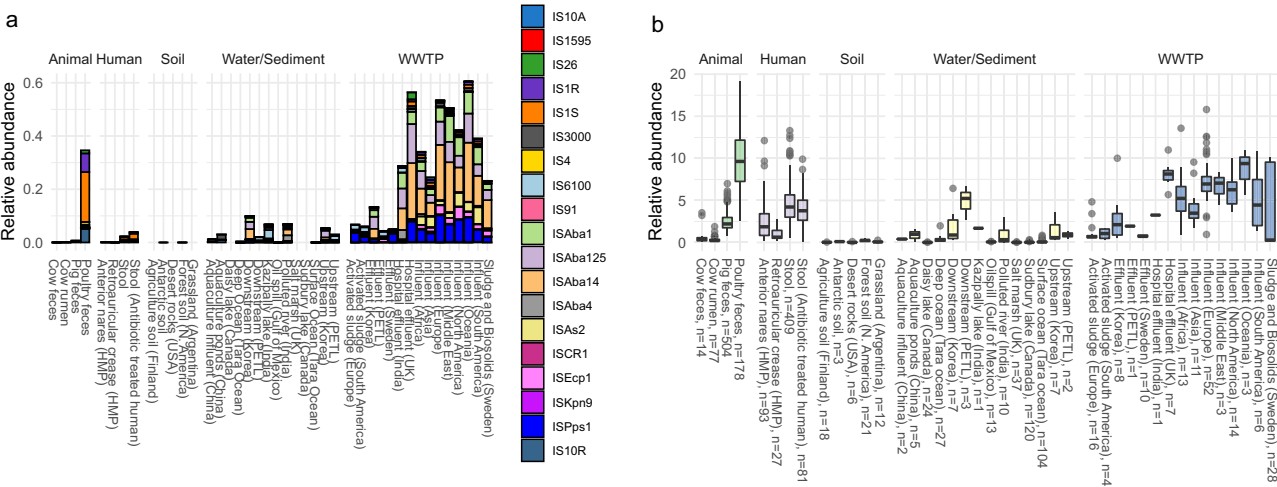

**Fig. 3 Relative abundance of IS elements.** The average relative abundance of specific IS elements likely to have been involved in the mobilization of an antibiotic resistance gene (MISE) (**a**) and relative abundance of all IS elements (**b**). Boxplot: A box represents the first (Q1) and third (Q3) quartile (the 25–75th percentile), the centerline represents the median. The dots represent values larger or smaller than Q1−1.5 or Q3 + 1.5 times the inter-quartile range.

**Co-existing origin species and MISE.** Selection pressure and ecological connectivity are two factors that are thought to play important roles for mobilization and emergence of ARGs[24,30]. Therefore, as a proxy for fecal contamination, and thus co-occurrence with species adapted to the human intestinal microbiota, we investigated the abundance of crAssphage in the selected environments. The levels of detected crAssphage were, unsurprisingly, the highest in human stool, untreated hospital effluent and WWTP influent (Supplementary Fig. 7). In addition, relatively high levels were detected in Pune river, upstream WWTPs in Korea, upstream PETL and in a salt marsh in the UK. When comparing the abundance of origin species and MISE in the soil and water environments combined with ($n = 22$) and without ($n = 391$) detected crAssphage, all species but *Shewanella algae* and all studied MISE were more abundant in the samples where crAssphage was detected and the same was true when only water environments with ($n = 21$) and without ($n = 332$) detected crAssphage were included (Wilcoxon-Mann-Whitney test, $p < 0.01$). Furthermore, an analysis of the fraction of samples that had detectable levels of both an origin species and its associated MISE showed that the proportion of samples containing complete pairs was much higher in the combined soil and water environments where crAssphage was detected (Fig. 4b and Supplementary Data 2) compared to the combined soil and water samples where no crAssphase was detected (Fig. 4a and Supplementary Data 2). This difference became even more prominent when comparing fractions of samples containing both origin species and corresponding MISE(s) in the WWTP influent with the stool samples (Fig. 4c, d and Supplementary Data 2). Here, the majority of the origin species-MISE pairs were present in the majority of the samples from the WWTP influent, while this was not the case for the human stool samples.

## Discussion

By analyzing a large range of public metagenomes, we show that almost all known origin species for mobile ARGs are considerably more abundant in wastewaters (and to some extent in contaminated water/sediments, in particular those suspected to have been contaminated by human feces) than in any other investigated environment type. While this also holds true in comparison with human microbiota, a small fraction of stool samples have a very high relative abundance of a few origin species. Most of the

origin species lacked the corresponding MISE in all analyzed genomes. This suggests that additional, potentially foreign, genetic material had to be present for the ARG to acquire intracellular mobility. Strikingly, those MISE are also most abundant in influents to WWTPs, while considerably less common in e.g., human stool metagenomes, where very few cases of co-occurrences between origin species and the corresponding MISE were found. This provides new evidence for wastewaters and WWTPs as plausible environments where ARGs were mobilized in the past and where additional ARGs could be mobilized in the future.

Almost all identified origin species have at some point been connected to infections in humans or animals. Therefore, it has been hypothesized that the corresponding ARGs were mobilized in the human and/or animal microbiota likely under selection pressure from antibiotics[15]. Here, we found that the average relative abundance of origin species was lower in the human microbiota compared to other environments, with the only exception being *Hafnia paralvei* that was on average more abundant in human microbiota, although not significantly so. Most other origin species were considerably less abundant, but still present in more than 5% of the human samples. The presence of *Citrobacter freundii*, *Hafnia alvei/paralvei*, *Klebsiella pneumonia* and *Morganella morganii*, is rather expected as they are known human commensals[31–34]. Occurrence of the origin species in a small proportion of humans and/or a relatively low abundance compared to wastewater environments likely contributes to a lower probability for mobilization and subsequent transfer of the corresponding ARG to pathogens. Still, it does not exclude that the human microbiome could have been the site where the mobilization took place. While many origin species are not permanent members of the human microbiome, an extrapolation of our results suggests that large numbers of people on earth still carry them, possibly as transient members of their intestinal microbiome. This might be sufficient for successful mobilization to occur. On the other hand, the fraction of human intestines where both the presence of a transient origin species, the corresponding MISE (also transient in most cases) and an antibiotic selection pressure coincide is likely very small. This should be compared with WWTP influents where the investigated origin species and corresponding MISE almost always co-occur, together with a mixture of antibiotics.

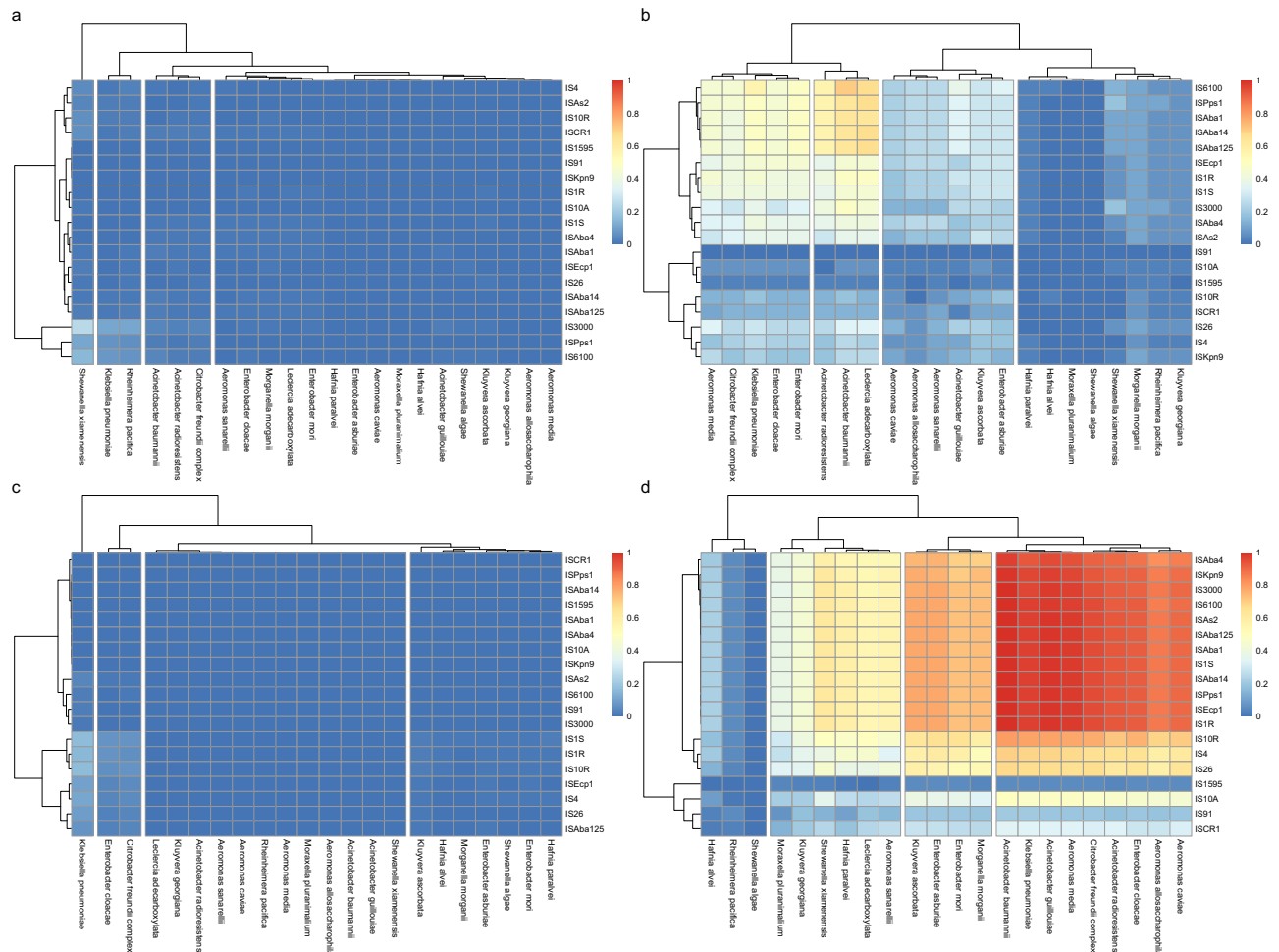

**Fig. 4 Co-existing origin species and mobilizing IS elements.** Fractions of samples that contained both an origin species and an IS elements involved in mobilization (MISE) for the following subsets of data, (**a**) combined soil and water environments where no crAssphage was detected ($n = 391$), (**b**) combined soil and water environments with detectable levels of crAssphage ($n = 22$), (**c**) human stool ($n = 384$), (**d**) influent of wastewater treatment plants ($n = 102$).

Antibiotic selection pressure could lead to an increase in abundance of some of the origin species since they carry an ARG. Strong selection pressures from antibiotics are more common in humans (and domestic animals) than in the external environment[35,36]. Still, wastewaters often harbor a mix of antibiotics in variable concentrations and, importantly, the selection pressure would therefore be more or less continuous[37]. Furthermore, the levels of antibiotics in discharges from antibiotic manufacturing can be staggering[38,39]. Indeed, we found high abundances of many of the investigated origin species in waters polluted with such industrial waste (e.g., Kazipally lake and upstream PETL where illegal dumping of industrial waste is the likely reason for high antibiotic concentrations). A recent study showed that exposure to untreated hospital effluent and WWTP influent from Sweden selects for multi-resistant *Escherichia coli*[40]. Such selection is likely even stronger in countries where the antibiotic consumption is higher than in Sweden[41,42]. While selection and enrichment of donors potentially contribute to the risk for mobilization and transfer, the most important role of antibiotic selection pressure is still likely the advantage it can provide to those strains where ARGs are more highly expressed. Although not all ARGs are sufficiently expressed in their original host to provide resistance[43–46], co-localization with IS-elements could increase the expression, and those strains of other species that eventually have received a mobilized ARG through

horizontal gene transfer could potentially benefit from this in an environment with antibiotic selection pressure[8,24].

It is known that WWTPs have an influx of human- and sometimes animal-related bacteria, (with some being pathogenic) as well as environmental bacteria[47,48]. Thus, it is not unexpected that we in sewage find origin species that are not detected in a limited set of human microbiota samples ($n = 1101$), and that we find origin species present in both human microbiota and in sewage[49]. Nevertheless, we did not expect the high degree of difference in abundance of origin species in WWTP influents relative to other environments, which could not easily be explained by high abundance in feces of a small proportion of the human population. All known origin species belonged to the phylum Proteobacteria. Therefore, environments where this phylum is dominating for any reason, could naturally have a higher relative abundance of the origin species. The WWTP environments investigated in this study were indeed dominated by Proteobacteria, but so were other environments such as water/ sediment and some soil samples (Supplementary Fig. 1), largely disproving this simplistic hypothesis. Instead, one of the most prominent differences between environments with a high relative abundance of origin species and those that had a low relative abundance seemed to be the exposure to human fecal matter. Indeed, the environments with the, on average, highest relative abundance of most origin species were also the environments

where we detected the highest levels of crAssphage, an indicator of human fecal pollution[27,50]. We also showed that many of the origin species are at least transient in the human microbiome. Thus, environments exposed to human fecal matter will have an influx of origin species, although in relatively low abundance. However, some common taxa in WWTP influents, such as *Aeromonas* spp., were more common in other environments than human stool, suggesting that for such taxa the input from fecal matter may be less important. It appears that several of the origin species are selected for in the WWTP or sewer environment, some potentially because of intrinsic resistance to antibiotics, but more likely because of other reasons. Indeed, it is known that the microbial composition of the WWTP gets altered throughout the treatment process, with many anaerobic species disappearing already in the sewage pipes[51–53]. In addition, we have not taken the natural lifestyle of bacteria such as biofilms into consideration. However, regardless of the reason for a higher relative abundance of origin species, the chances of a mobilization event taking place will inevitably increase.

Our analysis of the origin species' genomes showed that in most cases the origin species do not carry the corresponding MISE. However, it should be noted that although the corresponding MISE did not exist in the analyzed genomes it is still possible that they exist in some strains of those species, and for the ten origin species where the number of available complete genomes was lower than five the risk that we have missed the presence in a not yet sequenced strain is higher. Nevertheless, it appears that for the mobilization from many of the origin species to have occurred, it would have required the MISE to be provided by its surrounding environment. Among the studied MISEs, many were found to be more common in WWTP influents and hospital effluents compared to other environments, including the human microbiota. This could be due to the selection of those mobile genetic elements containing the MISE since many of them are strongly associated with different mobile ARGs. It could also be due to the greater variety of proteobacterial species in wastewaters compared to the human gut since proteobacterial species were shown to be overrepresented carriers of MISE[25,26]. Regardless, the origin species and the corresponding MISE coexisted in the majority of individual samples from WWTP-related environments, and in many environmental samples impacted by human feces. Human stool contained a relatively high abundance of IS elements, but a relatively low abundance of the studied MISE. Still, a few of the latter were found in large parts of the population, such as *Klebsiella pneumonia* (IS1 and IS10) and *Enterobacter cloacae* (IS4).

While the importance of selection pressures from antibiotics in the mobilization of ARGs has been discussed previously in the literature[54], we have here investigated the necessary presence and co-existence of origin species and MISE, and identified environments where this occurs. Still, there are numerous additional factors that can be of importance. We have touched upon the genetic compatibility between origin species and strains carrying MISE and concluded that they often seem to be relatively closely related. Still, both genetic and ecological factors (both biotic and abiotic) that differ between environments but are not known or investigated here are likely to influence the probability for successful mobilization of ARGs. The results in this study should therefore be interpreted as evidence, and not proof, for wastewaters as a likely site for mobilization events. Furthermore, environments where mobilization might have occurred does not necessarily reveal where future mobilization events will take place nor if the same MISE will be involved. Many of the species identified as origin species are well-studied species commonly associated with humans and animals[32,55–58]. Hence, it is not unexpected that we also detect those species in environments that

also are well studied[59]. However, the origins of the large majority of ARGs are still unknown, most likely because the species are not sequenced or not even discovered yet, alternatively that the mobilization event did not occur recently on an evolutionary scale. Nevertheless, the environments identified as potential places where mobilization events have happened might, based on their attributes, give us an indication of what kind of environments that have the potential to be the place where further mobilization events can take place[60].

In this study we have chosen to optimize a Kraken2 database for the identification of origin species. The optimization had a particular focus on limiting the number of false positives, thus, the sensitivity for finding some species was very low and it is very likely that we have missed the presence of some of these species in some environments where they are less abundant. However, as the purpose of this study was to compare environments against each other, we do not believe that this has affected the overall conclusions of this study. It should be reinforced that with the approach taken, it is not possible to compare the abundance of different species with each other because of the highly variable fraction of fragments classified to the species level. The low fraction for some species, such as *Klebsiella pneumonia*, was likely the result of the existence of very similar species such as *K. quasipneumoniae* and *K. variicola*. Furthermore, although we used the genomes of all species within the same genus as the corresponding origin species to improve and evaluate the database, it is possible that there are other species not yet sequenced that are similar to some of the origin species, and these could therefore have been misclassified. However, the estimation of the FPR based on the CAMI datasets showed that based on known genomes, very few reads were falsely classified as belonging to any of the origin species and we feel confident that the overwhelming majority of reads classified in this study are indeed correct.

The human and animal intestine have previously been proposed as likely environments for the mobilization and transfer of ARGs to pathogens[15]. Though this remains a possibility, we conclude that new knowledge presented in this study on the abundance of origin species and the associated mobilizing elements in different environments have led to more lines of evidence pointing toward wastewaters as the site for mobilization of known ARGs, compared to what was previously known. A growing amount of evidence show that several ARGs have been mobilized from the chromosome of their respective origin species repeatedly[61–63], indicating that mobilization is driven by non-random factors. Through a consistent combination of origin species, MISE and antibiotic selection pressure, wastewaters worldwide would provide a suitable environment for such repeated mobilizations. Although it is impossible to pinpoint where individual future mobilization events that will impact our ability to treat pathogens will occur, it is not unlikely that the identified environments could also be the arena for future mobilization events[60]. The present study shows that it is not enough to focus on surveilling and limiting known problematic species and ARGs, but in order to manage novel threats, more efforts are needed to understand the species, mechanism, drivers and environments involved in resistance gene mobilization. Wastewaters and polluted waterways may require some particular focus. Proper wastewater management may therefore not only reduce risks for environmental transmission of various pathogens, it may also reduce the risks for emergence of new resistance threats. Still, as there seem to be particularly good opportunities for mobilization and gene transfer already in influents of WWTPs (raw sewage), it is not self-evident how to mitigate risks beyond ensuring effective reduction of bacterial release to the environment.

## Methods

**Database creation and evaluation.** Recently Ebmeyer et al. published a study summarizing all known and verified taxa from which ARGs have been mobilized, i.e., when an ARG have transitioned from a highly immobile context on a chromosome to being easier transferrable through the connection with mobile genetic elements[15]. For some ARGs only the genera from which the ARG was mobilized from has been identified, but there is clear evidence for 23 unique species as being recent origins of around 30 ARGs (origin species). In many cases there are also strong indications on certain IS-elements being involved in the mobilization process (MISE) from those species. Based on the availability of sequenced genomes of the confirmed origin species, we chose 22 to be included in this study (Table 1).

The main objective of this study relied on the ability to compare relative abundances in environments where the origin species exist rather than quantifying their absolute abundance. Therefore, it was more crucial to use a method that would give as few false positives as possible, than to optimize for a low level of false negatives. Our choice of method was to use Kraken2, a k-mer based approach operating directly on short reads, which has previously been shown to efficiently handle large amounts of data[64], together with a custom made database. The custom database was specifically designed and optimized to identify the species investigated in this study while limiting the number of species, especially those being closely related to the investigated ones, of being falsely classified as an origin species. In essence, only discriminatory regions of the DNA are used for species classification, while those that overlap between closely related species are not used.

The custom database was created by first downloading the Kraken2 databases archaea, bacteria, human, plasmid, UniVec_Core and viral using the "kraken2-build–download-library" command (downloaded in December 2020). The bacterial database was then parsed to check which of the 22 investigated origin species, listed in Table 1, that were already included. The species that were not included as genomes in the original Kraken2 database were retrieved from NCBI assembly (https://www.ncbi.nlm.nih.gov/assembly) as complete genomes, and if no complete genome existed the representative genome was retrieved. Next, all representative genomes of all species within the same genus as their respective origin species were, if such genomes existed, retrieved from the NCBI assembly database (Supplementary Data 1). As plasmids can be exchanged between species, and often contain many mobile genetic elements, all sequences annotated as plasmids were extracted from all these sequences and their taxonomic id were changed to represent the category "unidentified plasmid", this was done to limit the number of fragments matching a plasmid as being inaccurately classified as belonging to a certain species. A Kraken2 database was built using the commands "kraken2-build–add-to-library" and "kraken2-build–build".

To optimize the ability of the method to distinguish reads from the origin species from reads of other species of the same genera, all genomes of the origin species that were not included to build the database were downloaded. Here, genomes that were marked complete and had the NCBI taxonomy status "ok" were downloaded. All genomes of the origin species, both in and not in the database, were then used to simulate paired-end reads of length 100 and 150 base-pairs using art_illumina, parameters "-l 100/150 > -m > 300/350 > -s 0 -na -p -f 1 -ss HS25"[65]. The same procedure was then applied on all genomes of the same genus as their respective origin species. Kraken2 was run on all simulated reads using the command "kraken2 –paired –confidence <confidence_value >" with the value of the parameter confidence ranging from 0 to 1 with 0.1 increments. The result was inspected and the value of the parameter confidence was decided to be set to 0.3. This value was chosen to minimize the number of fragments of species from the same genus as the origin species being falsely classified as origin species and did come with a cost of a lower detection rate of the origin species. The FPR, defined as proportion of fragments falsely classified as an origin species out of all fragments classified at a species level, was then estimated on low, medium and high complexity datasets generated by the Critical Assessment of Metagenome Interpretation (CAMI) consortium[66], and generated an average FPR of $9.59 \times 10^{-7}$ (Supplementary Table 2).

A final evaluation of the fraction of true and false positives was done in March 2021 when new genomes, not included in our database, of origin species and species of the same genera as the origin species had been added to NCBI GenBank. Here Kraken2 was run using the command "kraken2 –paired –confidence 0.3". The average TPR, defined as the fraction of correctly identified fragments out of all fragments classified at a species level, was 0.97 for reads of length 100 and 150 bp (Supplementary Table 1). For fragments from genomes not present in the database, the corresponding number was 0.92. However, the proportion of fragments that could be assigned at the species level was, on average, 0.56 and 0.43, for fragments from genomes present and not present in the database, respectively, reflecting the amount of fragments being discarded based on uncertainty. As a worst-case scenario, we tested the ability to not falsely classify fragments as origin species when the fragments were generated from genomes of the same genus as the origin species, with none of the tested genomes being included in the database. Here the average FPR was $1.36 \times 10^{-2}$ and $1.48 \times 10^{-2}$ with an average fraction of fragments classified at species level of 0.30 and 0.29 for read lengths of 100 and 150 bp respectively (Supplementary Table 3).

**Datasets and raw metagenomic analysis.** The data analyzed in this study consisted of publicly available metagenomic datasets generated from non-amplified short read data (Table 2 and Supplementary Table 4). The data was downloaded during the autumn of 2020 and was chosen to represent various environments within the main categories human, animal, water, soil and WWTPs and only samples with a minimum of 20 million fragments were included. The data was processed using an in-house built pipeline that consisted of the following analysis. First, the data was quality controlled and adapter sequences were removed using BBduk v. 38.86 (BBMap software)[67]. The resulting reads were counted and basic statistics were gathered using seqtk v. 1.3-r115-dirty, parameter "fqchk"[68]. Then the data was analyzed using Kraken2 v. 2.0.9 together with the custom made database, parameter "kraken2 –paired–confidence 0.3". Then, the data was searched for IS elements using an in house made database[25] consisting of transposase sequences identified from literature and NCBI, manually annotated against the ISFinder database[69], using diamond, parameters "blastx–max-target-seqs 3–query-cover 50–id 80". Finally, the data was searched for the presence of crAssphage. Here the crAssphage genome (NC_024711.1) was downloaded from GenBank and indexed using bowtie2-build[70]. The reads were then aligned to the genome using bowtie2 parameters "-x -1 < read1 > -2 < read2 >".

**Post-processing and analysis.** The results from the raw analysis were parsed in python and gathered as count tables. For the data analyzed for IS elements, only hits with an identity higher than 98 percent were kept. Metadata from all datasets were downloaded from their respective bioproject record in NCBI. The count tables together with the metadata was loaded into R and phyloseq objects were created using the phyloseq package (v.1.30.0) where the taxonomical counts were normalized with the total number of reads classified as bacteria in each sample[71]. The IS element counts were normalized by the length of their respective gene and the total number of million fragments for each sample. For the IS elements, only genes with non-zero counts in at least 2.5% of the samples per dataset were kept. To further reduce the noise between samples, the majority of the data was aggregated based on shared attributes (pig gut, human stool etc.) and the average for each attribute were then computed. The barplots were generated using ggplot2 v. 3.3.0 and the abundance heatmaps were generated using plot_heatmap from phyloseq while the prevalence heatmaps were generated using pheatmap from the pheatmap package v. 1.0.12[72]. The rarefed data used to estimate diversity, presence and to calculate correlation tables was generated using rrarefy from the vegan package v. 2.5.6[73]. For the taxonomical data (i.e., species and phyla counts) each sample was rarefied to contain $5*10^5$ bacterial fragments and the IS-count data was rarefied to contain $10*10^6$ fragments. A species was said to be present if at least 10 fragments were classified as belonging to the corresponding species, while an IS element was said to be present if the length normalized count was higher than 0.001. The diversity was estimated using the function diversity from the vegan package.

**Statistics and reproducibility.** For each origin species and origin IS, the difference in abundance between environments was assessed using two-sided Wilcoxon-Mann-Whitney tests, and false positives were controlled based on the false discovery rate (FDR) estimated by the Benjamini-Hochberg algorithm. The difference of origin species and origin IS being detected in environments were assessed on binary presence data (detected/not-detected), generated from the rarefed data sets, using Fisher's exact test and false positives were controlled based on the FDR estimated by the Benjamini-Hochberg algorithm.

**Reporting summary.** Further information on research design is available in the Nature Portfolio Reporting Summary linked to this article.

## Data availability

The data supporting the conclusions of this paper are publicly available at the European Nucleotide Archive and GenBank. All accession numbers are specified in Supplementary Table 4 and Supplementary Data 1. The source data behind the figures can be found in Supplementary Data 2. The custom-made database is available upon request.

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

## Acknowledgements

This work was funded by the Swedish Research Council VR grant numbers 2022-00945, 2018-05771 and 2018-02835, and the Swedish Research Council FORMAS grant number 2018-00787. Open access funding was provided by the University of Gothenburg.

## Author contributions

F.B., S.E., E.K., and D.G.J.L. designed the study. F.B. and S.E. collected the genomic data. F.B. created and validated the database. F.B. collected and curated the metagenomic data. F.B. created and ran the analysis pipeline. F.B., S.E., E.K., and D.G.J.L. analyzed the results. F.B. and D.G.J.L. drafted the paper. All authors discussed the results and its implications. All authors edited and approved the final paper.

## Competing interests

The authors declare no competing interests.
