## [Peer Review File · Communications Biology]

Reviewers' comments:

Reviewer #1 (Remarks to the Author):

The study entitled "Evidence for wastewaters as environments where mobile antibiotic resistance genes emerge" aims to demonstrate that wastewater is a privileged environment where antibiotic resistance genes can be acquired by novel hosts. The manuscript refers to an interesting data set that describes the co-occurrence of abundant bacterial groups and genetic elements involved in recombination events in wastewater environments. The study is a continuation of a previous publication by the same group - *Commun Biol* 4, 8 (2021).

The narrative of this submission elaborates on the acquisition and mobilization of novel antibiotic resistance genes based on a probabilistic rationale. Important ecology and evolution concepts are largely neglected. Despite the merits of the approach used, the results presented are not conclusive and may lead to incorrect interpretations. Some aspects should be (re)considered:

1. It is assumed that members of a given species will have the same fitness in all environments where they can thrive (e.g. wastewater or human stool). This is incorrect. Besides other factors may be strongly influenced by the microbial community composition, density (cells per volume or area), exogenous stress factors, among many others. Another assumption that may be not straightforward is that different strains of a given genus (or species) will have identical fitness under a specific conditions set.

2. It is assumed that if some taxa and some genetic elements prevail in a given environment (co-occurrence), they are likely associated. Although this is a logical principle, it is not necessarily true, and therefore should not be tacitly accepted unless evidence of such association is demonstrated. However, co-occurrence is the strongest argument provided.

3. It is assumed that the process of acquisition of novel antibiotic resistance genes involves the most abundant bacterial groups in a given environment. If there is any evidence that demonstrates this, it should be clearly provided in the manuscript. Stochastic events are probably part of the resistance acquisition process. Also, the spacial distribution of donor and recipient bacteria may be important in this process and it cannot be predicted based on abundance. Hence, it may be erroneous to generalize that acquisition of novel resistance genes will involve the most abundant species or a specific type of environment.

Other comments:

It is difficult to understand how the authors identified the "origin species", which criteria or information sources were used? – this is not for the reader.

L11- The question may be not where, but when and how antibiotic resistance is acquired for the first time.

L39 – "origin species" is misleading; indeed it seems the authors are referring to founder effects and founder species;

The hypothesis of the study is not totally clearly stated and aligned with experimental design;

L108 – which species hold these MISE? Which is the average MISE number per bacterial genome? How different is that average number in different environments? This information might be useful for a better understanding of the data and the results.

L115 and following – Since the methods are the last section of the paper, there should be a better description of procedures in the results section, to allow the reader to follow a rationale. For instance, it would be beneficial to have a brief explanation of the database and the optimization procedure, the criterion to select "origin species", etc..

L117 - It seems that this validation may have some bias, since with genomes the probability of unspecific identification is much lower than in a metagenome data set. Was this normalized taking such bias into account?

Most of the MISE presumably associated with antibiotic resistance genes and IS (figure 3) are indeed observed in raw influent. Although the number of metagenomes analysed is small, it seems that the relative abundance of these genetic elements is reduced during wastewater treatment. According with

these data, and to support the hypothesis raised in the title of manuscript, which would be the preferential niche for antibiotic resistance acquisition? The plumbing system?
In general, there is an important bias for proteobacteria in wastewater, so the conclusions are strongly influenced by this fact. It is assumed that because wastewater have higher abundance of Proteobacteria than other environments, they are probably acquiring novel genes in that environment. Although this is a logical argument, it seems to lack strong evidence. When wastewater is compared with human stool, it might be more accurate if one considers the ratio "origin species": members of the same phylum in that environment. Otherwise, all data is strongly biased by abundance.

Reviewer #2 (Remarks to the Author):

Summary

The authors use a set of species to look for them in public metagenomic data from different environments. The metagenome analysis shows that wastewaters and human stools have a high abundance of the set of species. They analyze some ISs in the set of species and find that IS content is the highest in hospital effluent, poultry faeces and WWTP. Their results suggest that residual water could be a site for the mobilization of antibiotic-resistance genes. However, there are important issues that need to be addressed before this study can be considered for publication.

Major comments

1. There is an important piece of data or analysis missing. The authors did not present the data where one can see that the ISs are carrying antibiotic resistance genes (ARGs). The authors conducted analyses showing the presence or absence of the ISs in the origin species but it's not shown if these ISs have ARGs. Equally important, the authors basically just analyze ISs and not a good diversity of Mobile Genetic Elements (i.e. plasmids, phages, pathogenicity islands, etc.). Considering this issue, the title and abstract are misleading. Given what is stated in the introduction, specifically lines 40-47, I assume they have already conducted the analysis linking ISs to ARGs in the "origin" species in a previous study, if so please say that explicitly.
2. Why did the authors choose the species listed in Table 1 as their origin species? Please give the reason behind it. There are some ESKAPE pathogens but also some other bacterial species that their clinical relevance is not obvious. Again, I think the authors used the set of species considered in a previous study by them.
3. For reproducibility sake, the accession number (or BioSample number) should be provided for all the genomes downloaded from the NCBI assembly site – this can be added a supplementary table. Please, also mention in the text how many genomes were included for each origin species. This is important to know how much information was included for each origin species.
4. The use of the word origin (of ARGs) is sometimes elusive. For instance in line 81. Do you mean the origin of a new allelic variant? Clearly, you do not mean the origin of the whole gene families. Please define more precisely what you mean by origin. Maybe you want to say the "most recent origin".

Minor comments

The discussion is rather long and, in some places, rather redundant.

Phages have also shown to be agents mobilizing ARGs, see references below

<https://pubmed.ncbi.nlm.nih.gov/32109174/>

<https://pubmed.ncbi.nlm.nih.gov/34232073/>

Lines 454, 465 and 394: should be genus (the singular of genera).

Lines 455-457: why did you do that with the sequence annotated as plasmids?

Lines 462-465: Please provide more details about the simulated reads for reproducibility's sake.

The coverage for the IS identification ("—query-cover 50") is a bit low. I would suggest at least 60% or even 70% to be sure that you covering most of the gene and not just 50%

Please provide references for ggplot2, phyloseq and vegan.

Reviewer #4 (Remarks to the Author):

The authors present a novel bioinformatic analysis approach that was developed to identify environments that are likely the origin (or hot spots) for the mobilization of known ARGs that were not mobile before. The authors used a customized database that is composed of origin species for specific ARGs that were non mobile and a priori present on the chromosome of these organisms, and later mobilized by the association of insertion sequences. The authors developed an approach to identify and define the origin species in a previous study, which is essential to read to understand the concept and the analysis approach.

Overall, the authors present very interesting and novel findings, by identifying and highlighting wastewater environments as putative hotspots for past and future mobilization events for ARGs. This was achieved by detecting the abundance of the defined origin species and their association to mobilizing insertion sequences (MISE) in metagenomic datasets from wastewater compared to soil, water, human and animal samples. The approach is novel. The authors did also correlate the presence of origin species and MISE with the human faecal marker crAssphage in their datasets. Surprisingly, a high correlation to origin species and MISE pairs was found for the presence of crAssphage in all samples but human stool.

Overall, the article needs revision to present the approach and results in a more comprehensive way. The discussion needs to be revised, and some details to be clarified, e.g. references or specific selection criteria for their ARGs and MISE (apart from their own reference).

Furthermore, additional analysis to put their approach into the context of current literature (as proposed below) could increase the value of their findings.

My overall advice would be a major revision.

Specific comments/ suggestions:

Introduction

Line 50/51: reference 16: to my understanding this reference makes no claim supporting this statement. This article highlights that the development of methicillin resistance predates the clinical use of beta-lactams, not that the origin species evolved so far that it cannot be identified anymore. Maybe the authors could rephrase and explain what they meant by this statement- or whether their statement is an interpretation of the results / study they are referring to.

Lines 69 to 79 are based on 2 review papers and the authors hypothesis only. It does also not consider natural lifestyle of bacteria (biofilms), and the role of free and external DNA and naturally competent cells in complex environments (neither in the environment studied)

The first paragraph of the results section is very difficult to understand and seems to fit better into the methods section.

The definition of origin species is not clear and requires the reader to fully study the previously published work by the same authors (reference 15). I would suggest to re-phrase this paragraph and to explain how the origin species are defined. The optimized database should be explained (which criteria) and referenced. The authors do not describe clearly how small fragments of 100 to 150bp from metagenomic sequence data are enough to identify as the origin species (what type of fragments? Filtered 16S rRNA gene fragments? Or several reads mapping on different part of the genome of the origin species? How is that specific enough to allow the identification of a species based on small reads? It could be a fragment that originates from another species (the core genome for example) or genus closely related). It would be great if the authors could explain how they identified / classified short metagenomic sequencing reads on species level. It seems like a truly innovative and valuable approach; however, it is very difficult to grasp (also when reading their other paper in which they curated the database of origin species) the way it is written. I would suggest an extended method section reintroducing the method, the theory, the definition of origin species. Technical and statistical details as to validating their data by using TPR and FPR should be in supplementary data. Figure 1 could use some clarifications for example how many origin species are represented (all? Is that the full database).

The fact that the two Indian sites contained the highest amount of origin species is interesting – would that not indicate that rather the pollution/ selection level of antibiotics is determining the abundance of those species not the environment per se? Hence the fact that wastewater contain chemical and pharmaceutical pollutants might explain why these origin species are e.g. higher in WW than in the human gut on average for example? This would also correlate with findings in lines 179-185.

Figure 2 – the authors specified that they used all genomes available for the origin species originally identified to carry non mobile ARGs – meaning for 10 species there were less than 5 genomes available for this analysis? Is it safe to assume that because the IS elements were not present on those genomes that they were not present before or in other isolates of the respective species (not sequenced or not included into analysis)?

Line 262 to 267 and Figure 4: It is not clear which fraction of the presented data are water or soil samples. Are they mixed? Or does one-part of the figure 4 a and b represent water and the other part soil samples? Please clarify.

Interesting finding that MISE and origin species pairs were associated with crAssphage in all environments but the human gut? Any explanation?

Discussion

The discussion states the most important findings and provides a hypothesis to explain these findings. The discussion should be shortened and focus more on the recent literature of other groups. In general, the discussion lacks references for certain claims (lines: 309, 310; 339; 349,350; 374, 375,405, 406;416; 416-419).

Line 362: Do the authors mean treated hospital wastewater by hospital effluent? And WWTP influent refers to untreated, urban wastewater?

Lines 383-399 discusses the authors implemented approach. This should rather go to the beginning or the end of the discussion- and as mentioned before also partly into the results and methods section so the reader can understand the approach (without having to fully study their previous paper).

Lines 400 to 419 are redundant of what has been concluded in the text before.

Concerning some aspects that are missing in the discussion:

For example, a recently published paper by Che et al., <https://doi.org/10.1073/pnas.2008731118> suggested that a general evolutionary mechanism for the horizontal transfer of AMR genes is mediated by the interaction between conjugative plasmids and ISs.

Are the most relevant identified ISs in this study similar / the same as the studied MISE here?

The authors could consider including the identified conjugative plasmids and ISs by Che et al, in their analysis pipeline. This could be a very important addition and a means to compare different analysis approaches to see whether results are comparable- and extend the findings of Che et al., to the large environmental metagenomic datasets used here.

The fact that the wastewater environment is on average and in a more permanent way an environment under high selective pressure containing chemical, pharmaceutical, heavy metals, micropollutants and surfactants, than the human gut which is only sporadically exposed to e.g. high doses of antibiotics or other chemicals/pharmaceuticals should be considered and discussed in the context of current literature.

How important is the presence of the origin species in a certain environment compared to the presence of ISs or MISEs (the latter seems much more relevant)

R1:

The study entitled "Evidence for wastewaters as environments where mobile antibiotic resistance genes emerge" aims to demonstrate that wastewater is a privileged environment where antibiotic resistance genes can be acquired by novel hosts. The manuscript refers to an interesting data set that describes the co-occurrence of abundant bacterial groups and genetic elements involved in recombination events in wastewater environments. The study is a continuation of a previous publication by the same group - Commun Biol 4, 8 (2021).

The narrative of this submission elaborates on the acquisition and mobilization of novel antibiotic resistance genes based on a probabilistic rationale. Important ecology and evolution concepts are largely neglected. Despite the merits of the approach used, the results presented are not conclusive and may lead to incorrect interpretations. Some aspects should be (re)considered:

1. It assumed that members of a given species will have the same fitness in all environments where they can thrive (e.g. wastewater or human stool). This is incorrect. Beside other factors may be strongly influenced by the microbial community composition, density (cells per volume or area), exogenous stress factors, among many others. Another assumption that may be not straightforward is that different strains of a given genus (or species) will have identical fitness under a specific conditions set.

Authors' reply: It is, indeed, true that fitness for different members of different species varies across different environments but, in contrary to what the reviewer says, we do not make any assumptions regarding fitness in this manuscript. Our assumption is that a higher abundance of donor species and corresponding mobilizing elements (given everything else is similar) will increase the chance of a mobilization event to have taken/to take place. And as we *observe* high abundances of many donor species and mobilizing elements in e.g. sewage, we do not rely on theoretical assumption about their fitness.

We do, however, discuss *why* the abundance of certain species is higher in certain environments (see e.g. paragraph 2, 3 and 4 in the discussion), and certainly many factors indeed can affect the fitness.

This comment is related to comment 3, so please also see answer to comment 3.

2. It is assumed that if some taxa and some genetic elements prevail in a given environment (co-occurrence), they are likely associated. Although this is a logical principle, it is not necessarily true, and therefore should not be tacitly accepted unless evidence of such association is demonstrated. However, co-occurrence is the strongest argument provided.

Authors' reply: We do not make any assumption that a given taxa and specific mobile genetic elements are genetically coupled/associated. From the metagenomic data, we do not have the possibility to determine if the fragments we identify belong to the same cell or not. What we do is that we investigate if they are present in the same physical environment (not necessarily in the same cell), and if they are, we assume a higher possibility of them to become genetically associated. The opposite is apparent; if they are not found in the same physical environments, certainly chances for becoming genetically associated are lower.

3. It is assumed that the process of acquisition of novel antibiotic resistance genes involves the most abundant bacterial groups in a given environment. If there is any evidence that demonstrates this, it should be clearly provided in the manuscript. Stochastic events are probably part of the resistance acquisition process. Also, the spacial distribution of donor and recipient bacteria may be important in this process and it cannot be predicted based on abundance. Hence, it may be erroneous to generalize that acquisition of novel resistance genes will involve the most abundant species or a specific type of environment.

Authors' reply: In contrast to what the reviewer says, we do not assume or state that the process of acquisition of novel ARGs involves the most abundant bacterial groups in a given environment. What we have done in this study is that we have compared the abundance of 22 species that already have been shown to be the recent origin species for certain ARGs, in various environments. We are not comparing the abundance of one species to that of another. We only compare one species at a time across environments to see where they are more or less abundant (or even absent).

What is correct is that we assume that a higher abundance of an origin species in one environment compared to another would lead to more chances for mobilization to have occurred through stochastic processes (given everything else is similar). It should however be pointed out that this was only one of the factors we considered. We also determined where the investigated origin species and the mobilizing genetic elements involved in mobilization of ARGs from the investigated species were present and abundant (i.e. above/below detection limit) – since both the species and corresponding mobilizing elements are needed in order for mobilization to have occurred, and environments where this criterium is not fulfilled simply could not have been where mobilization occurred (given that the condition in the environments stays the same). There has also been new evidence pointing towards that several ARGs have been repeatedly mobilized from the chromosome of their origin species, which would speak in favour of an environment where these conditions are constantly (or at least regularly) fulfilled as the site of mobilization. We have added this argument in the last paragraph of the discussion (lines 438-442).

The reviewer is correct that the spatial distribution within a given environment can also play a role (probably more so in solid media such as soils, versus liquid media such as wastewaters where bacteria move more freely), but that does not make abundance unimportant. The same thing goes for selection pressure from antibiotics, which we also discuss in the manuscript (paragraph 2, 3 and to some extent 4).

We have given some thought to why the reviewer seems to have misinterpreted our study on several points, and based on comments given by reviewer 4 we have substantially extended and added information about our method and the background of our study (i.e. the origin species). See lines 39-49 in introduction, a new 1st paragraph in Results as well as a new 1st paragraph in Methods, together with more clarifications throughout the text. We have also removed the (previously) 4th paragraph of the introduction where we describe a later part of the chain (i.e. horizontal gene transfer), as we suspect that this paragraph added confusion about the aim of the study.

Other comments:

It is difficult to understand how the authors identified the "origin species", which criteria or information sources were used? – this is not for the reader.

Authors' reply: We suspect that many comments given by both reviewer 1 and 2 are based on misinterpretations of our study and the concept of origin species. The might partly be because they have not thoroughly read the current manuscript and the study by Ebmeyer et al., which this study is based on, but also because we have been unclear. We have therefore extended the background information about the concept of mobilization and origin species with lines 29, 39-48 in the introduction, a new first section in results as well as a new first section in the methods.

For the reviewer: The origin species we investigated in this study have not been established by us in more than in a few cases – they have been discovered by many different research groups but were summarized in a paper by us in 2021 (Ebmeyer et al., 2021). This should now be more clearly stated in many places in the manuscript.

L11- The question may be not where, but when and how antibiotic resistance is acquired for the first time.

Authors' reply: We do not agree with the reviewer. If we don't know *where* resistance is acquired for the first time we risk on focusing our mitigation efforts on the wrong environments. Furthermore, one does not exclude the other. While the information on *how* and *when* antibiotic resistance is acquired also is important, the focus of this study is "*where*".

L39 – "origin species" is misleading; indeed it seems the authors are referring to founder effects and founder species;

Authors' reply: As mentioned above, we believe that the reviewer has misunderstood the concept of this study. We are very aware of the concept of founder effects in ecology, but what we investigate has nothing to do with founder effects. As mentioned in the comment above, we have now clarified the concept of origin species on several locations in the manuscript.

The hypothesis of the study is not totally clearly stated and aligned with experimental design;

Authors' reply: The aims of the study are clearly stated in the last paragraph of the introduction. We believe that if one understands the concept of origin species and MISE, the aims of the study will become clear. Therefore we have, as mentioned above, added more background and explanations of the concept and the study. We also removed a paragraph (previously paragraph 4) in the introduction which we believe caused confusion regarding the aim.

L108 – which species hold these MISE? Which is the average MISE number per bacterial genome?

Authors' reply: We thank the reviewer for the comment and have, based on the reviewer's suggestion, searched all genomes present in NCBI RefSeq for the MISE. We found only 223 unique species that carried a MISE (there were 7283 unique species present in RefSeq), and that Proteobacterial species were overrepresented carriers of MISE. Furthermore, we found that in general, species carrying MISE were often more closely related to the origin species in which the MISE was associated with mobilization than to other origin species. We have extended the results section with lines 234-245 and supplementary figure 6 to describe these results.

How different is that average number in different environments? This information might be useful for a better understanding of the data and the results.

Authors' reply: Since the data we analysed consisted of metagenomic fragments we cannot give a reliable estimate of the number of MISE per carrier, instead we refer the reviewer to figure 3 which shows the average relative abundance of MISE in each environment (fig 3a) and relative abundance of all IS-elements in each environment (fig 3b).

L115 and following – Since the methods are the last section of the paper, there should be a better description of procedures in the results section, to allow the reader to follow a rationale. For instance, it would be beneficial to have a brief explanation of the database and the optimization procedure, the criterion to select “origin species”, etc..

Authors' reply: We thank the reviewer for the comment and have, as mentioned above, extended the introduction and added new paragraphs to both results and methods. With regards to “selecting origin species” we did include all known origin species with complete sequenced genomes (specified in Table 1) that fulfil the criteria with regards to sufficient evidence, as lined out in Ebmeyer et al (2021).

L117 - It seems that this validation may have some bias, since with genomes the probability of unspecific identification is much lower than in a metagenome data set. Was this normalized taking such bias into account?

Authors' reply: The evaluation of the method was done using simulated fragmented data from known genomes. This is necessary since we need to have full knowledge about the content of the simulated data in order to estimate the error rates. We have now clarified this on several locations in the text (lines 131-132,134, 137). See also the results from the validation on complex metagenomic samples generated by CAMISIM (supplementary table 2).

Most of the MISE presumably associated with antibiotic resistance genes and IS (figure 3) are indeed observed in raw influent. Although the number of metagenomes analysed is small,

Authors' reply: We believe that the reviewer is referring to the number of included studies, not the number of metagenomes. The number of metagenomes included in our analysis is 2496 (totalling more than 22 trillion bases). This is a very substantial number given the current standards of the field (although there are some studies that are even larger).

it seems that the relative abundance of these genetic elements is reduced during wastewater treatment. According with these data, and to support the hypothesis raised in the title of manuscript, which would be the preferential niche for antibiotic resistance acquisition? The plumbing system?

Authors' reply: Yes, this is correct. We have in many places throughout the manuscript changed the word WWTP to wastewaters.

In general, there is an important bias for proteobacteria in wastewater, so the conclusions are strongly influenced by this fact. It is assumed that because wastewater have higher abundance of Proteobacteria than other environments, they are probably acquiring novel genes in that environment. Although this is a logical argument, it seems to lack strong evidence. When wastewater is compared with human stool, it might be more accurate if one considers the ratio “origin species”: members of the same phylum in that environment. Otherwise, all data is strongly biased by abundance.

Authors' reply: What is interesting is the relative abundance of origin species (and MISE), not the taxonomic distribution of all the bystanders in the communities. The reasons for why the abundance of an origin species/MISE might be higher in one environment than another may overlap with reasons for why abundance of proteobacteria in general is higher. However, and this is very important, *why* the origin species/MISE are more abundant in one environment compared to another does not matter for the conclusion on the probability that they will be able to interact and become genetically associated. We simply hypothesize that presence and high abundance of origin species/MISE in a given environment is to some extent linked to higher risk for them to interact (given everything else alike).

We would again also like to clarify that we are not comparing abundances of all species (only the investigated origin species) and we do not, in contrast to what the reviewer says, assume that Proteobacteria is acquiring novel genes in wastewaters. We do, however, hypothesize that the origin species, which all are proteobacteria, might have acquired MISE in wastewaters.

R2:

Summary

The authors use a set of species to look for them in public metagenomic data from different environments. The metagenome analysis shows that wastewaters and human stools have a high abundance of the set of species. They analyze some ISs in the set of species and find that IS content is the highest in hospital effluent, poultry faeces and WWTP. Their results suggest that residual water could be a site for the mobilization of antibiotic-resistance genes. However, there are important issues that need to be addressed before this study can be considered for publication.

Major comments

1. There is an important piece of data or analysis missing. The authors did not present the data where one can see that the ISs are carrying antibiotic resistance genes (ARGs). The authors conducted analyses showing the presence or absence of the ISs in the origin species but it's not shown if these ISs have ARGs. Equally important, the authors basically just analyze ISs and not a good diversity of Mobile Genetic Elements (i.e. plasmids, phages, pathogenicity islands, etc.). Considering this issue, the title and abstract are misleading. Given what is stated in the introduction, specifically lines 40-47, I assume they have already conducted the analysis linking ISs to ARGs in the "origin" species in a previous study, if so please say that explicitly.

Authors' reply: Unfortunately, the reviewer seems to have largely misinterpreted the rationale of this study. We investigate already identified origin species, identified as origins of ARGs in previous studies and outlined by Ebmeyer et al. (Ebmeyer et al., 2021). The mobilization of ARG(s) from these species have in many cases been associated (in previous studies by other researchers) with certain IS-elements. We are therefore not interested in if the investigated IS-elements carry an ARG or not, but merely look for their presence, since they would likely have been required to be in the same physical environment as the origin species for the mobilization of the respective ARG to have happened. We are analysing only IS-elements because we are investigating the 22 origin species from which there is already published evidence that mobilization of specific ARGs likely has happened with the help of specific IS-elements.

Since we have noted that both reviewer 1 and reviewer 2 have misunderstood the concept of origin species and therefore also the concept of the study, we have removed one potentially confusing

paragraph in the introduction as well as expanded the introduction, results and methods with new paragraphs and explanations about origin species and our aims with this study. See also answers to reviewer 1.

2. Why did the authors choose the species listed in Table 1 as their origin species? Please give the reason behind it. There are some ESKAPE pathogens but also some other bacterial species that their clinical relevance is not obvious. Again, I think the authors used the set of species considered in a previous study by them.

Authors' reply: We have included all species with available complete genomes that have been shown, in previous studies largely by other authors, to be the recent evolutionarily origin of different ARGs. The list of all previously confirmed origin species, and the evidence for these being the origin of specific ARGs, can be found in a published article by Ebmeyer et al. (Ebmeyer et al., 2021) where the current knowledge within the field was summarized. We have clarified this on several locations in the manuscript. See also answers to comment 1 and reviewer 1.

3. For reproducibility sake, the accession number (or BioSample number) should be provided for all the genomes downloaded from the NCBI assembly site – this can be added a supplementary table. Please, also mention in the text how many genomes were included for each origin species. This is important to know how much information was included for each origin species.

Authors' reply: We have now added supplementary data containing the accession number for the downloaded genomes as well as number of genomes included for each origin species and we are referencing this data on lines 123, 224 and 487. Since we are analysing 22 origin species, we do not, for the readability of the text, want to write out the number of included genomes for each species in the text but instead reference to the supplementary data. Though we now have included the number of genomes for each origin species, we argue that the extra value gained from knowing exactly how many genomes of the origin species that were included is limited. Instead, the results from the validation of the method contain detailed information regarding number of genomes tested, both in and not in the database together with the estimates of the method's performance.

4. The use of the word origin (of ARGs) is sometimes elusive. For instance in line 81. Do you mean the origin of a new allelic variant? Clearly, you do not mean the origin of the whole gene families. Please define more precisely what you mean by origin. Maybe you want to say the "most recent origin".

Authors' reply: The definition of gene families is different for different ARG classes, and the names of ARGs does not always, unfortunately, follow the definition for a gene family of the respective class (Bush & Jacoby, 2010; Jacoby et al., 2008). However, if one classifies a group of ARGs with the same name as the same gene family, we do in some instances mean whole gene families (see (Ebmeyer et al., 2021)). For instance, the beta-lactamase "gene family" FOX, (including all mobile allelic variants in that group, i.e. FOX-1, FOX-2, FOX-3 and so on) has been recently mobilized from *Aeromonas allosaccharophila*, while the "gene family" SHV was mobilized from *Klebsiella pneumoniae* etc. However, since the definition of gene family is varying and is not always coherent with the names of the ARGs, we have abstained from introducing the concept in the manuscripts since we are a) not investigating the ARGs themselves, and b) we believe it will only confuse the reader.

Instead we refer the reader to Ebmeyer et al. to read more about the different ARGs with a known origin species and we have added lines 39-48 where we explain the definition and concept of origin species. We have also changed the wording in line 79 to make it more coherent with our previous

and later use of “origin species”.

Minor comments

The discussion is rather long and, in some places, rather redundant.

Authors’ reply: We agree with the reviewer and have removed the whole second to last paragraph, which contained a lot of repetition of what had already been said.

Phages have also shown to be agents mobilizing ARGs, see references below

<https://pubmed.ncbi.nlm.nih.gov/32109174/>

<https://pubmed.ncbi.nlm.nih.gov/34232073/>

Authors’ reply: This study is focusing on the 22 origin species from which certain ARGs most likely have been recently mobilized with the help of IS-elements. Although phages can be involved in the horizontal transfer of ARGs, we have not yet seen conclusive evidence pointing to a single ARG which has been recently mobilized from a defined taxa with the help of phage(s). Of course, this doesn’t exclude that phages might have mobilized ARGs (making a previously largely immobile chromosomal ARG more easily movable), but we are using a list of confirmed origin species. The first article referenced above describes predicted prophages in *A. baumannii* genomes, and conclude that many of the predicted prophages encode ARGs. However, they cannot determine if they contribute to ARG spread, and even less so if the finding is related to mobilization of ARGs. The second article referenced aimed to investigate the distribution of ARGs and virulence factor genes within prophages in seven pathogens. Here they discover that some prophages can harbour many ARGs and that these ARGs were located near mobile genetic elements. Again, although it is possible that phages play a role in horizontal transfer of ARGs and/or mobilization of ARGs, none of the provided articles show any conclusive evidence of mobilization.

Lines 454, 465 and 394: should be genus (the singular of genera).

Authors’ reply: We thank the reviewer for pointing this out and have made changes accordingly.

Lines 455-457: why did you do that with the sequence annotated as plasmids?

Authors’ reply: Since most plasmids can be easily exchanged between species we cannot accurately assign fragments belonging to a plasmid to a certain species. We have added lines 487-491 to motivate our decision to create a separate plasmid category.

Lines 462-465: Please provide more details about the simulated reads for reproducibility’s sake.

Authors’ reply: We thank the reviewer for pointing this out and have added more details (line 498).

The coverage for the IS identification (“—query-cover 50”) is a bit low. I would suggest at least 60% or even 70% to be sure that you covering most of the gene and not just 50%

Authors’ reply: Query coverage refers to the coverage of the sequences which are compared to the database (which contains the subject sequences), in our case the query sequences therefore are the

metagenomic fragments (and not as the reviewer suggest, the genes). Since the shortest transposases within the IS-elements in our reference database are around 80bp, and our fragments range from 75-150 bp, our query-coverage was set so that the longest fragments (i.e. 150 bp) would have a chance to pass our criteria for the shortest transposases. If we would set the query cut-off to 60%, then we would require 90 bp of our 150bp long fragment to match, and this would be impossible for the shorter transposases.

Please provide references for ggplot2, phyloseq and vegan.

Authors' reply: We thank the reviewer for pointing this out and have added the references.

Reviewer #4 (Remarks to the Author):

The authors present a novel bioinformatic analysis approach that was developed to identify environments that are likely the origin (or hot spots) for the mobilization of known ARGs that were not mobile before. The authors used a customized database that is composed of origin species for specific ARGs that were non mobile and a priori present on the chromosome of these organisms, and later mobilized by the association of insertion sequences. The authors developed an approach to identify and define the origin species in a previous study, which is essential to read to understand the concept and the analysis approach.

Overall, the authors present very interesting and novel findings, by identifying and highlighting wastewater environments as putative hotspots for past and future mobilization events for ARGs.

This was achieved by detecting the abundance of the defined origin species and their association to mobilizing insertion sequences (MISE) in metagenomic datasets from wastewater compared to soil, water, human and animal samples. The approach is novel. The authors did also correlate the presence of origin species and MISE with the human faecal marker crAssphage in their datasets. Surprisingly, a high correlation to origin species and MISE pairs was found for the presence of crAssphage in all samples but human stool.

Overall, the article needs revision to present the approach and results in a more comprehensive way.

The discussion needs to be revised, and some details to be clarified, e.g. references or specific selection criteria for their ARGs and MISE (apart from their own reference).

Furthermore, additional analysis to put their approach into the context of current literature (as proposed below) could increase the value of their findings.

My overall advice would be a major revision.

We thank reviewer 4 for a good description of our study, and for recognizing its merits.

Specific comments/ suggestions:

Introduction

Line 50/51: reference 16: to my understanding this reference makes no claim supporting this statement. This article highlights that the development of methicillin resistance predates the clinical use of beta-lactams, not that the origin species evolved so far that it cannot be identified anymore. Maybe the authors could rephrase and explain what they meant by this statement- or whether their statement is an interpretation of the results / study they are referring to.

Authors' reply: The reviewer is correct that the statement on line 51-52 was an interpretation of the cited research. Indeed, to show that an ARG has been mobilized from a not presently existing species would be very difficult, if not impossible. We have therefore removed the reference, but are keeping the conceptual statement which still holds (line 57).

Lines 69 to 79 are based on 2 review papers and the authors hypothesis only. It does also not consider natural lifestyle of bacteria (biofilms), and the role of free and external DNA and naturally competent cells in complex environments (neither in the environment studied)

Authors' reply: We have completely removed the paragraph in which the referenced lines were located. We did this because in this paragraph we were describing a later step of the chain, i.e. HGT, and not when an ARG initially becomes mobile (associated with an IS-element), and we believe that this was one of the reasons that other reviewers misinterpreted the aims of our study. We have however modified the article on other places to also include free and external DNA as a potential source of MISE, see lines 101, 244. In addition, we have previously taken the possibility of external DNA into consideration when discussing the initial association (see lines 308 and 390).

It is true that other factors such as biofilms could affect the potential for a mobilization and/or transfer of an ARG. However, we have in this study chosen to focus on the environments on a larger scale such as sewers, soil, waters etc, and not considered the microenvironments within these larger categories. We have added a sentence in the discussion to clarify that biofilms is an aspect we have not considered in our analysis (lines 381-382).

The first paragraph of the results section is very difficult to understand and seems to fit better into the methods section.

The definition of origin species is not clear and requires the reader to fully study the previously published work by the same authors (reference 15). I would suggest to re-phrase this paragraph and to explain how the origin species are defined.

Authors' reply: We thank the reviewer for this comment and have written a new first paragraph of the results section explaining on what criteria we chose the origin species included in this study. This paragraph also includes a non-technical description about the database creation. We have also re-written the original first paragraph describing the results from the database evaluation. In addition, we have extended the second paragraph of the introduction with more information about the concept of origin species, including a clear definition.

The optimized database should be explained (which criteria) and referenced.

Authors' reply: In addition to the newly written first paragraph of the Results section (see answer above), the database creation and evaluation is explained in paragraph 2, 3 and 4 in the Methods

section. We have extended all these paragraphs with more details and explanations (see answer below). The result from the evaluation is available in supplementary table 1, 2, and 3. In addition, we have added a statement in “data availability” saying that the database is available upon request.

The authors do not describe clearly how small fragments of 100 to 150bp from metagenomic sequence data are enough to identify as the origin species (what type of fragments? Filtered 16S rRNA gene fragments?)

Authors’ reply: We thank the reviewer for pointing this out. The foundation of the method is the software Kraken2 with optimized parameters, together with a custom-made database optimized for the identification of the 22 origin species we investigate. Kraken2 operates directly on reads using a k-mer based approach, i.e. not just 16S rRNA gene fragments (Wood et al., 2019, p. 2). To explain this, we have added a line in the newly written first paragraph of the results (line 120) as well as added a sentence on lines 471-477 in Methods.

Or several reads mapping on different part of the genome of the origin species? How is that specific enough to allow the identification of a species based on small reads? It could be a fragment that originates from another species (the core genome for example) or genus closely related). It would be great if the authors could explain how they identified / classified short metagenomic sequencing reads on species level. It seems like a truly innovative and valuable approach; however, it is very difficult to grasp (also when reading their other paper in which they curated the database of origin species) the way it is written.

Authors’ reply: The reviewer is correct in that it is infamously difficult to classify short reads at species level. That is why we have created a custom database optimized to correctly classify reads originating from specifically the 22 investigated origin species, while not classifying reads from other species, especially from closely related genera, as belonging to any of our origin species (i.e., the database is not optimized to correctly identify other species). We extended the default Kraken2 database (containing bacterial, human, plasmid, known vector and viral genomes) with many manually confirmed genomes from genera closely related to each origin species, in order to avoid false classification of reads coming from these. In essence, that means that only discriminatory regions of the DNA are used for classification, while those that overlap between closely related species are not used for classification to the species level. In addition, we chose the “confidence” parameter of kraken2 to be 0.3 (default is 0). Simplified, this means that if uncertainty is too high, the read will not be classified at species level. Therefore we have a low proportion of reads being assigned at all at species level (see supplementary table 2 and 3), as, for this study, we decided it is better to discard data than to falsely classify it as an origin species. It is therefore important to note that we cannot measure absolute abundance of an origin species in a certain environment, but we can compare the abundances within species across samples.

In addition, the method was rigorously tested. As the reviewer pointed out, one would suspect that reads coming from a closely related genus will have the highest risk of being falsely classified as an origin species, especially if this species was not included in the database (i.e. we have no knowledge about this genome). Therefore, one of our evaluation strategies was to test the method on fragmented genomes from closely related genera, where the genomes used for testing were not included in our database (see supplementary table 3). The method was also tested on fragmented genomes from our 22 origin species (genomes both included and not included in the database) as well as complex metagenomic samples generated by Critical Assessment of Metagenome Interpretation (CAMI) consortium (Sczyrba et al., 2017).

This is now further explained in the first and second paragraph of results, and we have extended paragraph 2 and 3 of the method section. In addition, the performance of the method is discussed in paragraph 7 of the Discussion section.

I would suggest an extended method section reintroducing the method, the theory, the definition of origin species.

Authors' reply: We thank the reviewer for this comment and have added a new paragraph in the beginning of the method section where we again introduce the concept of origin species and mobilizing IS elements as well as our rationale for choosing the 22 origin species investigated in this study. In addition we have extended paragraph 2 and 3 to include more details and explaining concepts about the method creation and evaluation.

Technical and statistical details as to validating their data by using TPR and FPR should be in supplementary data.

Authors' reply: We have shortened the paragraph in Results which reports the TPR and FPR (second paragraph) by removing details. However, we believe that it is of importance to keep the summarised results of the evaluation as they give confidence to the method on which many of the results in the article rely on.

Figure 1 could use some clarifications for example how many origin species are represented (all? Is that the full database).

Authors' reply: We have modified the legend of Figure 1 to clarify that it shows results for the 22 investigated origin species. The database contains many more bacteria (and other genomes, see answer above), but it is optimized for the 22 origin species.

The fact that the two Indian sites contained the highest amount of origin species is interesting – would that not indicate that rather the pollution/ selection level of antibiotics is determining the abundance of those species not the environment per se?

Authors' reply: Indeed, it is a possible explanation that a high antibiotic load on an environment could lead to a higher relative abundance of certain origin species. We discuss this hypothesis in paragraph three and to some extent paragraph four in the discussion. In addition, we have added lines in paragraph three to further lift the finding of the high levels of origin species in environments polluted with waste from drug manufacturing (line 341-344) and a clarification that not all ARGs are sufficiently expressed in their original host (i.e. the origin species) (lines 350-352).

Hence the fact that wastewater contain chemical and pharmaceutical pollutants might explain why these origin species are e.g. higher in WW than in the human gut on average for example? This would also correlate with findings in lines 179-185.

Authors' reply: Indeed, we did see a significantly higher abundance of *Enterobacter mori* in humans treated recently treated with antibiotics compared with the ones that had not taken antibiotics recently. We saw the same tendency for *Morganella morganii*, although the difference was not significant. However, the antibiotic pollution/exposure does not seem to be the complete answer as the abundance of all investigated origin species was higher in wastewaters and/or polluted waters environments compared to the antibiotic-treated humans, indicating that other factors may play an even larger role. Furthermore, since only one origin species out of 22 was significantly more abundant in antibiotic-treated humans compared to the non-antibiotic treated, strong antibiotic

selection pressure does not seem to significantly affect the presence/abundance of most origin species in the human gut (they need to be present to begin with).

Figure 2 – the authors specified that they used all genomes available for the origin species originally identified to carry non mobile ARGs – meaning for 10 species there were less than 5 genomes available for this analysis? Is it safe to assume that because the IS elements were not present on those genomes that they were not present before or in other isolates of the respective species (not sequenced or not included into analysis)?

Authors' reply: For the analysis of origin species carrying the MISE we wanted to use complete genomes so that we could separate plasmids from chromosomes. Therefore the reviewer is correct that for ten of the investigated species this analysis was only done on less than five genomes. We agree with the reviewer that the absence of MISE in the few genomes does not mean that they are not present in any other member of that species, although it at least shows that they are not present in the core genome of that species. However, the co-existence analysis of origin species and corresponding MISE still shows clear patterns, indicating that in many cases the origin species do not carry the corresponding MISE. We have added a sentence in discussion (lines 385-389) to point out that an absence of MISE in the analysed genomes, especially when few genomes were available, does not automatically mean that they are not present in any isolate.

Line 262 to 267 and Figure 4: It is not clear which fraction of the presented data are water or soil samples. Are they mixed? Or does one-part of the figure 4 a and b represent water and the other part soil samples? Please clarify.

Authors' reply: The presented data in Figure 4a and 4b is mixed soil and water samples with no detected crAssphage (4a) and detected crAssphage (4b). We thank the reviewer for the comment and have added a clarification on lines 281 and 287-289, as well as modified the legend of Figure 4.

Interesting finding that MISE and origin species pairs were associated with crAssphage in all environments but the human gut? Any explanation?

Authors' reply: Although we find a relatively high abundance of IS-elements in the human gut, the abundance of MISE was not as high as in e.g. wastewaters. One explanation for this could be the lower relative abundance of proteobacterial hosts of MISE in human gut (Supplementary figure 1), as the MISE seem to be more common in proteobacterial species compared to species from other phyla (see new added results described in line 234-245 and new supplementary figure 6). For the origin species, many of them seem to be transient in the human gut and they all belong to the phylum proteobacteria which seem to thrive especially in the water environments, while other bacterial taxa which may “dilute” the abundance of origin species in the human gut, do not. Furthermore, wastewater, and probably other environments contaminated with human feces, contain bacteria from many different individuals. Hence it is possible that a source of the investigated origin species in the environments where crAssphage was detected is indeed human feces, but while we don't detect the MISE and origin species pair in many samples in the human gut we do so it in environments contaminated with human feces. This is discussed in paragraph 4 and 5 in the Discussion section.

Discussion

The discussion states the most important findings and provides a hypothesis to explain these findings.

The discussion should be shortened and focus more on the recent literature of other groups.

Authors' reply: We have significantly shortened the discussion by completely removing the second to last paragraph. We have added references at suitable places, according to suggestions by the reviewer. However, many of the lines that the reviewer wanted to have references for were in the concluding second to last paragraph that has now been removed. See below.

In general, the discussion lacks references for certain claims (lines: 309, 310;

Authors' reply: The sentence on lines 328-330 (previous 309-312) is based on our results in this study (since no-one else has studied the distribution of the origin species in various environments). We have modified the sentence to make this clear.

339;

Authors' reply: The presented phylum distribution of the investigated environments is based on results obtained in this study (lines 156-160 in Results and Supplementary figure1). We have modified the sentence on line 364 (previously 339) to clarify this as well as added a reference to Supplementary figure 1.

349,350;

Authors' reply: We believe that the reviewer means the sentence "However, some common taxa in WWTP influents, such as *Aeromonas* spp, were more common in other environments than human stool, suggesting that for such taxa the input from fecal matter may be less important." As this is an interpretation of our own results we do not see how we could reference this.

374, 375, 405, 406; 416; 416-419).

Authors' reply: We have added references to claims made on the stated lines. Please note that the lines 406, 416 and 416-419 were located in the paragraph that we have removed.

Line 362: Do the authors mean treated hospital wastewater by hospital effluent? And WWTP influent refers to untreated, urban wastewater?

Authors' reply: We mean untreated hospital effluent, and have clarified this in Table 1. The reviewer is correct that we mean untreated urban wastewater, and we have added a clarification in Table 1 as well as changed the heading of paragraph 3 in Results to say wastewaters instead of WWTP. However, since we sometimes refer to the whole group "WWTP" we want to avoid confusion and continue to use WWTP influent when describing the results from this particular environment. We are using the word wastewaters in the introduction and discussion when talking about the results in general.

Lines 383-399 discusses the authors implemented approach. This should rather go to the beginning or the end of the discussion- and as mentioned before also partly into the results and methods section so the reader can understand the approach (without having to fully study their previous paper).

Authors' reply: We have now removed the second to last paragraph (previous lines 400-419) and the paragraph discussing the implemented approach is now the last paragraph before the conclusion. See above for the extension of Results and Methods with more descriptions about the method.

Lines 400 to 419 are redundant of what has been concluded in the text before.

Authors' reply: We thank the reviewer for this comment and we agree. We have therefore removed this paragraph.

Concerning some aspects that are missing in the discussion:

For example, a recently published paper by Che et al., <https://doi.org/10.1073/pnas.2008731118> suggested that a general evolutionary mechanism for the horizontal transfer of AMR genes is mediated by the interaction between conjugative plasmids and ISs.

Authors' reply: We have now (Introduction, lines 68-71) added a sentence which cites Che et al. (2021) together with a previously published study from our group (Razavi et al., 2020) also showing the association patterns of ARGs and IS-elements.

Are the most relevant identified ISs in this study similar / the same as the studied MISE here?

Authors' reply: As was already reported by Razavi et al. 2020, many of the in the study by Che et al. identified most relevant IS, are indeed the same as the studied MISE. As we in this study did not identify the MISE, (this has already been done by other researchers) we do not see the added value in discussing why many of the MISE often are associated with mobile ARGs, as it is out of scope (and discussed by others, see e.g. Razavi et al.). We have however added a sentence in the discussion, line 393, saying that many of the MISE are strongly associated with mobile ARGs, where we cite Razavi et al. and Che et al.

The authors could consider including the identified conjugative plasmids and ISs by Che et al, in their analysis pipeline. This could be a very important addition and a means to compare different analysis approaches to see whether results are comparable- and extend the findings of Che et al., to the large environmental metagenomic datasets used here.

Authors' reply: Although the drivers of horizontal transfer of ARGs is an important subject, that represents a later stage of the chain of events, and is out of scope for this study. Note that we here are focusing on the mobilization process, that is where a chromosomally fixated ARG gains the ability to move, rather than the actual HGT process of the ARGs. In addition, it would be very difficult, if not impossible, to do distinguish specific conjugative plasmids from fragmented metagenomic data with reasonable confidence (we are working with reads of the length 75-150 bp).

The fact that the wastewater environment is on average and in a more permanent way an environment under high selective pressure containing chemical, pharmaceutical, heavy metals, micropollutants and surfactants, than the human gut which is only sporadically exposed to e.g. high doses of antibiotics or other chemicals/pharmaceuticals should be considered and discussed in the context of current literature.

Authors' reply: Although we discuss this on lines 331-354 (end of paragraph 2 and paragraph 3), we have clarified that the selection pressure is likely more or less continuous in many wastewaters (line 338-340).

How important is the presence of the origin species in a certain environment compared to the presence of ISs or MISEs (the latter seems much more relevant).

Authors' reply: Since you need both the donor bacteria and the MISE for mobilization to have occurred, we argue that both are equally important.

Reviewers' comments:

Reviewer #1 (Remarks to the Author):

In my opinion, the problem of this study is that the authors are elaborating a bold conclusion about antibiotic resistance acquisition (not necessarily emergence) based solely on statistical evidence. The ecology and genetics of antibiotic resistance acquisition is rather complex and although statistical analysis may be essential to study the process it must be complemented by biological evidence (e.g. genetic linkage, host-mobile genetic element, other). As part of the rebuttal the authors say that "What we do is that we investigate if they are present in the same physical environment (not necessarily in the same cell), and if they are, we assume a higher possibility of them to become genetically associated." This is in my opinion the major drawback. The way the study is designed, technically correct but disregarding microbial ecology and genetics, may lead to incorrect conclusions. The rationale of the paper is based on co-occurrence, which is not evidence in the absence of other biological support.

Reviewer #4 (Remarks to the Author):

2nd review

General comments:

I thank the authors for addressing most of the reviewers' comments.

Have some additional observations and minor suggestions.

I state that the introduction is more fluid to read and that the main objective of the study became clearer.

Abstract:

line 13: instead of "majority of known species" I suggest "majority of previously identified species".

line 20: instead of "provide new evidence" write "highlight/ suggest WW as..."

Introduction:

Line 48: MISE is introduced the first time here and needs to be fully explained.

87-89—the sentence is not clear not clear . crAssphage is a marker for human faecal pollution -and high abundance of crAssphage was correlated to high levels with ARGs (Karkman 2019 Nat Com). Detecting crAssphage is hence usueful to that end – but I would not know what the authors mean with their sentence here – which risk factors?

96" well characterized " should be "previously describe"

Line 101 "by using a database optimized for this purpose" is not clear. Which purpose? And how was the database optimized and why? But in 1 or 2 sentence.

The authors gave a very clear and detailed response to my comment (explaining how they optimized the database)- that should be explained in the manuscript text, and even briefly in the introduction- perhaps just saying that "an analysis approach was implemented that allowed a specific search and subsequent classification of metagenomic sequencing data according to the origin species" (in a phrase before define origin species; in methods and results explain how you curated the database which optimized your analysis approach to the introduced purpose (your new analysis approach). I feel lots of unclarity is coming from to technical description, or repetitive description of methods and results— so the non-expert reader can not follow.

107: " our analysis provides new evidence pointing to " should be rather " our analysis points towards wastewaters as.."

To me the summary of the findings is still not clear:

MISE suspected to have been involved in mobilisation – is there proof? how? Is that from Ebmeyer et al. ?

Which reference shows that these MISE were associated with the ARGs of the origin species – or rather in their mobilisation—?

And how does the absence of these MISE justify the assumption that intracellular mobilisation of ARGs from origin species was achieved by acquisition of MISE from other species?

The authors need to be clearer and more precise in summarizing the main findings and message at the end of the introduction.

Results:

113: “performance of a method” should be “implementation of a method”

130-131 what does it mean to have genomes included and not included? This is confusing (move to methods, where its well exxplained, or see the authors response to my comments, also well explained- but remove from results)

131-148: this belongs to methods. Its very confusing to read this (even if relevant) in the results section

A general remark I think (as said above) methods are present too detailed throughout the manuscript—in a too technical language—which is why it is difficult to follow results, findings, and objectives.

Overall, the reviewers have addressed the comments in their rebuttal letter.

I think the manuscript could still be optimized by separating methods and method verification from results and the main findings.

Fine tuning of language will help to get the message across to the reader.

I think this is very interesting and important work- and as often, the authors are aware of all details and all steps during method validation and development – however, they are not essential for the reader to understand the objective, and main findings.

The authors have already improved the manuscript, and objectives are clearer, but they could still profit from clearer description and a more general description of main objectives and findings.

The methods are cleaner now, and the added results in SI help to understand the methods and results in depth.

I congratulate the authors on this additional work.

I would recommend a minor additional review for more clarity.

Reviewers' comments:

Reviewer #1 (Remarks to the Author):

In my opinion, the problem of this study is that the authors are elaborating a bold conclusion about antibiotic resistance acquisition (not necessarily emergence) based solely on statistical evidence. The ecology and genetics of antibiotic resistance acquisition is rather complex and although statistical analysis may be essential to study the process it must be complemented by biological evidence (e.g. genetic linkage, host-mobile genetic element, other). As part of the rebuttal the authors say that "What we do is that we investigate if they are present in the same physical environment (not necessarily in the same cell), and if they are, we assume a higher possibility of them to become genetically associated." This is in my opinion the major drawback. The way the study is designed, technically correct but disregarding microbial ecology and genetics, may lead to incorrect conclusions. The rationale of the paper is based on co-occurrence, which is not evidence in the absence of other biological support.

Authors' reply: We agree with the reviewer that solely the presence of an origin species and a corresponding MISE does not automatically mean that mobilization must have occurred/will occur in the environment. As pointed out previously, our argument is that environments where we have the two main elements required for mobilization (i.e. the origin species and the MISE) will have a significantly higher probability for mobilization events compared to environments where this is not fulfilled (the ARG cannot be mobilized if it is not present!). So, in some contrast to the reviewer's conclusion, we do believe that co-occurrence is indeed a piece of evidence for mobilization in a certain environment, but it is of course not proof that mobilization happened in the environment, which is an important distinction to make. As a comparison, traces of DNA on a crime scene from a suspected murderer is certainly considered "evidence", but not necessarily "proof" (unless backed up with other lines of evidence). We also never claim that mobilization happened/will happen in the identified environments, but point out that the likelihood for it happening is larger.

In addition, we would like to stress that we do not base our identification of wastewaters as plausible arenas for the mobilization of ARGs solely on statistical evidence, as the reviewer suggests. We do bring up an entirely new piece of evidence based on the coexistence of origin species and MISE. The presence of a more or less continuous selection pressure from a mixture of antibiotics and ecological connectivity, pieces of evidence that has been raised earlier, are also discussed in the manuscript. But of course, we acknowledge that there are still knowledge gaps to be filled, including for example evidence generated from controlled lab experiments repeating the history of mobilization under controlled conditions, reflecting those in different environments, including wastewaters.

We have rewritten the third to last paragraph of the discussion section, and as suggested by the editor, amending limitations to what extent one can and cannot make conclusions based on our findings. (note that limitations are also indicated in other parts of the discussion, e.g. lines 310, 364, 368-372, 407-410, 411-419). The start of the paragraph now reads:

"While the importance of selection pressures from antibiotics in the mobilization of ARGs has been discussed previously in the literature⁵⁴, we have here investigated the necessary presence and co-existence of origin species and MISE, and identified environments where this occurs. Still, there are numerous additional factors that can be of importance. We have touched upon the genetic compatibility between origin species and strains carrying MISE and concluded that they often seem to be reasonably closely related. Still, both genetic and ecological factors (both biotic and abiotic) that differ between environments but are not known or investigated here are likely to influence the probability for successful mobilization of ARGs. The results in this study should therefore be

interpreted as evidence, and not proof, for wastewaters as a likely site for mobilization events. Furthermore, environments where mobilization might have occurred does not necessarily reveal where future mobilization events will take place nor if the same MISE will be involved.”

Reviewer #4 (Remarks to the Author):

2nd review

General comments:

I thank the authors for addressing most of the reviewers’ comments.

Have some additional observations and minor suggestions.

I state that the introduction is more fluid to read and that the main objective of the study became clearer.

Abstract:

line 13: instead of “majority of known species” I suggest “majority of previously identified species”.

Authors’ reply: Changed to previously known origin species.

line 20: instead of “provide new evidence” write “highlight/ suggest WW as...”

Authors’ reply: While we firmly think the study provides new lines of evidence, we have rewritten this sentence, and it should be clear that there are remaining uncertainties. It now reads: “Our results identify wastewaters and wastewater-impacted environments as plausible arenas for the initial mobilization of resistance genes”

Introduction:

Line 48: MISE is introduced the first time here and needs to be fully explained.

Authors’ reply: We thank the reviewer for pointing this out and we have added an explanation.

87-89—the sentence is not clear not clear . crAssphage is a marker for human faecal pollution -and high abundance of crAssphage was correlated to high levels with ARGs (Karkman 2019 Nat Com). Detecting crAssphage is hence usefual to that end – but I would not know what the authors mean with their sentence here – which risk factors?

Authors’ reply: We understand the confusion and thank the reviewer for pointing it out. For a mobilized ARG to make its way to and emerge in a clinical setting, it would at some point need to transfer to bacteria that are compatible with the human microbiome. High abundance of crAssphage would indicate that the environment in question harbours a wide set of bacteria normally thriving in the human gut. The co-occurrence with such bacteria would resolve one of the limitations in the transfer chain from the environment to clinics. We have therefore modified the sentence slightly.

96” well characterized “ should be “previously describe”

Authors’ reply: Changed to previously described.

Line 101 “by using a database optimized for this purpose” is not clear. Which purpose? And how was

the database optimized and why? But in 1 or 2 sentence.

The authors gave a very clear and detailed response to my comment (explaining how they optimized the database)- that should be explained in the manuscript text, and even briefly in the introduction- perhaps just saying that “an analysis approach was implemented that allowed a specific search and subsequent classification of metagenomic sequencing data according to the origin species” (in a phrase before define origin species; in methods and results explain how you curated the database which optimized your analysis approach to the introduced purpose (your new analysis approach). I feel lots of unclarity is coming from to technical description, or repetitive description of methods and results— so the non-expert reader can not follow.

Authors’ reply: We have extended the sentence on line 101 to read “By using a database optimized for the purpose of discriminating reads from 22 origin species from that of other species...”. We have also moved paragraph 2 of the Results section to the Methods, so that the beginning of the Method section almost only focuses on a non-technical description of the method (lines 115-132). In addition, the method creation and evaluation are described in detail in paragraphs 2-5 (lines 484-516).

107: “ our analysis provides new evidence pointing to “ should be rather “ our analysis points towards wastewaters as..”

Authors’ reply: Changed to “points towards”.

To me the summary of the findings is still not clear:

MISE suspected to have been involved in mobilisation – is there proof? how? Is that from Ebmeyer et al. ?

Which reference shows that these MISE were associated with the ARGs of the origin species – or rather in their mobilisation—?

Authors’ reply: There is no proof that any particular MISE was involved in the mobilization of a particular ARG, there are however strong indications that they have been involved. Ebmeyer et al summarized all IS elements likely involved in the mobilization (together with the origin species and ARG), the information was gathered from multiple studies hence all references can be found in Ebmeyer et al. We describe this on lines 45-49 in the introduction, as well as on line 203 in results.

And how does the absence of these MISE justify the assumption that intracellular mobilisation of ARGs from origin species was achieved by acquisition of MISE from other species?

Authors’ reply: In some cases it could have come from the same species, however, the absence in all sequenced strains of a particular species still show that the MISE is at least not in the core genome. For those species where only a few strains are sequenced, this is a more likely possibility than for the species where there are sequenced genomes from a large variety of strains . We are leaving room for this possibility as we in the text write on line 99

“This indicates that in many cases the intracellular mobilization may have required the acquisition of MISE from another bacterial strain or species, either directly or indirectly through free DNA”.

And on line 369:

“However, it should be noted that although the corresponding MISE did not exist in the analyzed genomes it is still possible that they exist in some strains of those species, and for the ten origin species where the number of available complete genomes was lower than five the risk that we have missed the presence in a not yet sequenced strain is higher.”

The authors need to be clearer and more precise in summarizing the main findings and message at the end of the introduction.

Authors' reply: We hope that the changes made on lines 96, 101-102,104 and 108 in the final paragraph of the introduction have made the summary more clear.

Results:

113: "performance of a method" should be "implementation of a method"

Authors' reply: We have changed to "implementation of a method".

130-131 what does it mean to have genomes included and not included? This is confusing (move to methods, where its well exxplained, or see the authors response to my comments, also well explained- but remove from results)

Authors' reply: We agree with the reviewer that it is difficult to understand the included/not included in database without having read the methods. We have therefore taken the suggestion to move the whole paragraph (previously lines 129-148) to the Method section.

131-148: this belongs to methods. Its very confusing to read this (even if relevant) in the results section

Authors' reply: We have removed lines 129-148 and incorporated these results in the Method section.

A general remark I think (as said above) methods are present too detailed throughout the manuscript—in a too technical language—which is why it is difficult to follow results, findings, and objectives.

Authors' reply: We hope that the removal of the slightly technical description of the results from the evaluation of our method from the results section have made the Results easier to read. We have also removed one technical sentence on (previously) line 124, first paragraph of results. However, we believe that it is necessary to keep a high level of details in the Methods as other researchers should be able to know exactly what we have done and be able to recreate the results.

Overall, the reviewers have addressed the comments in their rebuttal letter.

Authors' reply: We thank the reviewer for noting this.

I think the manuscript could still be optimized by separating methods and method verification from results and the main findings.

Authors' reply: We have, as mentioned above, moved the paragraph about method evaluation from the Results section to the Methods section.

Fine tuning of language will help to get the message across to the reader.

Authors' reply: We have incorporated all changes suggested by the reviewer regarding the language, and hope that the message is now clearer.

I think this is very interesting and important work- and as often, the authors are aware of all details and all steps during method validation and development – however, they are not essential for the reader to understand the objective, and main findings.

Authors' reply: Although we previously have argued that the results from the method evaluation is part of the results, we agree with the reviewer that it makes the main findings of the study more difficult to understand. We have therefore removed the details of the method creation and evaluation from the results (as mentioned above), but chosen to keep a non-technical description of the foundation of the study (the origin species) and the method in the first paragraph of the results. This was suggested previously by the reviewers and we believe it enhances the understanding of the results.

The authors have already improved the manuscript, and objectives are clearer, but they could still profit from clearer description and a more general description of main objectives and findings.

Authors' reply: We hope that the changes made as suggested by the reviewer have made the main objectives and findings easier to understand.

The methods are cleaner now, and the added results in SI help to understand the methods and results in depth.

I congratulate the authors on this additional work.

I would recommend a minor additional review for more clarity.